RESEARCH COMMUNICATION

# Patronin governs minus-end-out orientation of dendritic microtubules to promote dendrite pruning in *Drosophila*

**Yan Wang[1,2†], Menglong Rui[1†], Quan Tang[1,3], Shufeng Bu[1,3], Fengwei Yu[1,2,3,4]\***

[1]Temasek Life Sciences Laboratory, National University of Singapore, Singapore, Singapore; [2]NUS Graduate School for Integrative Sciences and Engineering, Centre for Life Sciences, National University of Singapore, Singapore, Singapore; [3]Department of Biological Sciences, National University of Singapore, Singapore, Singapore; [4]Neuroscience and Behavioral Disorder Program, Duke-NUS Graduate Medical School, National University of Singapore, Singapore, Singapore

**\*For correspondence:**
fengwei@tll.org.sg

[†]These authors contributed equally to this work

**Competing interests:** The authors declare that no competing interests exist.

**Abstract** Class IV ddaC neurons specifically prune larval dendrites without affecting axons during *Drosophila* metamorphosis. ddaCs distribute the minus ends of microtubules (MTs) to dendrites but the plus ends to axons. However, a requirement of MT minus-end-binding proteins in dendrite-specific pruning remains completely unknown. Here, we identified Patronin, a minus-end-binding protein, for its crucial and dose-sensitive role in ddaC dendrite pruning. The CKK domain is important for Patronin's function in dendrite pruning. Moreover, we show that both *patronin* knockdown and overexpression resulted in a drastic decrease of MT minus ends and a concomitant increase of plus-end-out MTs in ddaC dendrites, suggesting that Patronin stabilizes dendritic minus-end-out MTs. Consistently, attenuation of Klp10A MT depolymerase in *patronin* mutant neurons significantly restored minus-end-out MTs in dendrites and thereby rescued dendrite-pruning defects. Thus, our study demonstrates that Patronin orients minus-end-out MT arrays in dendrites to promote dendrite-specific pruning mainly through antagonizing Klp10A activity.
**Editorial note:** This article has been through an editorial process in which the authors decide how to respond to the issues raised during peer review. The Reviewing Editor's assessment is that minor issues remain unresolved (see decision letter).
DOI: https://doi.org/10.7554/eLife.39964.001

## Introduction

In the developing nervous systems, neurons often undergo remodeling events, such as apoptosis, pruning and regrowth, which are pivotal for the refinement of mature neural circuits in both vertebrates and invertebrates (*Luo and O'Leary, 2005*; *Riccomagno and Kolodkin, 2015*). Pruning is a developmental process referred to as selective removal of unwanted neurites, for example axons, dendrites, or synapses, without causing neuronal death. After pruning, neurons often continue to extend their axons or dendrites to form the mature and functional connections. In vertebrates, various cortical, hippocampal and motor neurons prune their excessive neurites and re-wire into mature circuits. Defects in neuronal pruning result in larger dendritic spine density in layer V pyramidal neurons of autism spectrum disorder (ASD) patients (*Tang et al., 2014*). In invertebrates, such as *Drosophila*, the nervous systems undergo large-scale remodeling during metamorphosis, a transition from larval to adult stage (*Yu and Schuldiner, 2014*; *Kanamori et al., 2015*). In the central nervous system (CNS), mushroom body (MB) γ neurons prune their dorsal and medial axon branches as well as entire dendrites, and subsequently re-extend the medial branches to be part of the adult-specific circuits (*Lee et al., 1999*). In the peripheral nervous system (PNS), some dendritic arborization (da)

neurons, including class I da (ddaD/ddaE) and class IV da (C4da or ddaC) neurons, prune away all their larval dendrites but leave their axons intact (*Williams and Truman, 2005*; *Kuo et al., 2005*), whereas class III da neurons (ddaA/ddaF) are eliminated via apoptosis during the first day of metamorphosis (*Williams and Truman, 2005*). Pruning morphologically resembles axonal or dendritic degeneration following neurological diseases or injury. Thus, understanding the mechanisms of developmental pruning might provide some important insights into neurodegeneration in pathological conditions.

*Drosophila* C4da or ddaC neurons have been recognized as an appealing model system to unravel the mechanisms underlying dendrite-specific pruning during early metamorphosis. Upon induction of the steroid hormone 20-hydroxyecdysone (ecdysone) at late larval stage, ddaC dendrites are initially severed at their proximal regions as early as 4 hr after puparium formation (APF), followed by rapid fragmentation and debris clearance (*Figure 1A*) (*Williams and Truman, 2005*). Ecdysone binds to a heterodimeric receptor complex, which consists of the Ecdysone receptor and its co-receptor Ultraspiracle, to induce several downstream effectors, including a key transcription factor Sox14 (*Kirilly et al., 2009*; *Kirilly et al., 2011*), a cytoskeletal regulator Mical (*Kirilly et al., 2009*), Headcase (*Loncle and Williams, 2012*), a Cullin1 E3 ligase complex (*Wong et al., 2013*), and calcium signaling (*Kanamori et al., 2013*). Cytoskeletal disassembly, especially microtubule (MT) disassembly, is a key step of pruning occurring before the detachment of neurites in both MB γ and ddaC neurons (*Watts et al., 2003*; *Williams and Truman, 2005*). In ddaC neurons, MTs in the proximal dendrites break down prior to the dendritic membrane fission. Katanin p60-like 1 (Kat-60L1), a putative MT severing factor, is required for dendrite pruning of ddaC neurons (*Lee et al., 2009*), whereas other MT severing factors, including Katanin 60 (Kat-60), Spastin and Fidgetin, appear to be dispensable (*Lee et al., 2009*; *Stone et al., 2014*; *Tao et al., 2016*). In contrast, mammalian Spastin mediates branch-specific disassembly of MTs and facilitates synapse elimination in the developing mouse neuromuscular junctions (*Brill et al., 2016*). A recent study indicated that Par-1, whose mammalian ortholog is a Tau kinase, is required for increased MT dynamics and dendrite pruning in ddaC neurons. Par-1 promotes MT breakdown mainly via Tau inhibition (*Herzmann et al., 2017*).

MTs are polarized cytoskeletal structures that are assembled from heterodimers of α- and β-tubulin, and contain fast-growing plus ends and slow-growing minus ends (*Howard and Hyman, 2003*; *Akhmanova and Steinmetz, 2015*). Fast-growing plus ends, which are associated by plus-end-tracking proteins (+TIPs), are highly dynamic, whereas slow-growing minus ends are more stable with the decoration of minus-end-binding proteins (*Akhmanova and Steinmetz, 2015*; *Howard and Hyman, 2003*). In differentiated mammalian neurons, axons and dendrites differ in MT orientation: axonal MTs are mainly aligned in a plus-end-out pattern, whereas MTs are mixed at the proximal dendritic region with both minus-end-out and plus-end-out arrays (*Akhmanova and Steinmetz, 2015*). In *Drosophila* and *C. elegans* neurons, axons, like mammalian ones, contain nearly uniform plus-end-out MTs, however, MTs are oriented almost exclusively minus-end-out in major dendrites but mainly plus-end-out in shorter terminal dendrites (*Stone et al., 2008*; *Rolls et al., 2007*; *Ori-McKenney et al., 2012*; *Goodwin et al., 2012*; *Yalgin et al., 2015*). Growing studies have begun to unravel how the minus-end-out MTs are organized and maintained in the dendrites. In *C. elegans*, kinesin-1 was proposed to regulate predominant minus-end-out orientation of dendritic MTs through gliding plus-end-out MTs out of the dendrites (*Yan et al., 2013*). In *Drosophila*, minus-end-out orientation of dendritic MTs depends on +TIPs, plus-end-directed motor protein kinesin-1/2 as well as MT regulators γ-Tubulin/Centrosomin (*Mattie et al., 2010*; *Ori-McKenney et al., 2012*; *Nguyen et al., 2014*; *Herzmann et al., 2018*; *Yalgin et al., 2015*). However, whether and how minus-end-binding proteins regulate MT orientation in the dendrites remain elusive.

The Patronin family of proteins have been identified as conserved MT minus-end-binding proteins, which contain Patronin in *Drosophila*, calmodulin-regulated spectrin-associated protein 1/2/3 (CAMSAP1/2/3) in mammals, and PTRN-1 in worms (*Baines et al., 2009*). Patronin/CAMSAP/PTRN-1 contains a calponin homology (CH) domain at its amino-terminus, three predicted coiled-coil (CC) domains at its central region, and a CAMSAP/KIAA1078/KIA1543 (CKK) domain at its carboxyl-terminus (*Figure 3—figure supplement 1A*) (*Baines et al., 2009*). *patronin* (also known as *ssp4*) was first identified to regulate the length of mitotic spindle in *Drosophila* S2 cells (*Goshima et al., 2007*). The Patronin proteins recognize and stabilize free minus ends by protecting them from kinesin-13-mediated depolymerization (*Goodwin and Vale, 2010*). Mammalian CAMSAP3/Nezha anchors MT minus ends to adherence junctions and controls the apical-basal MT orientation in cultured epithelial cells

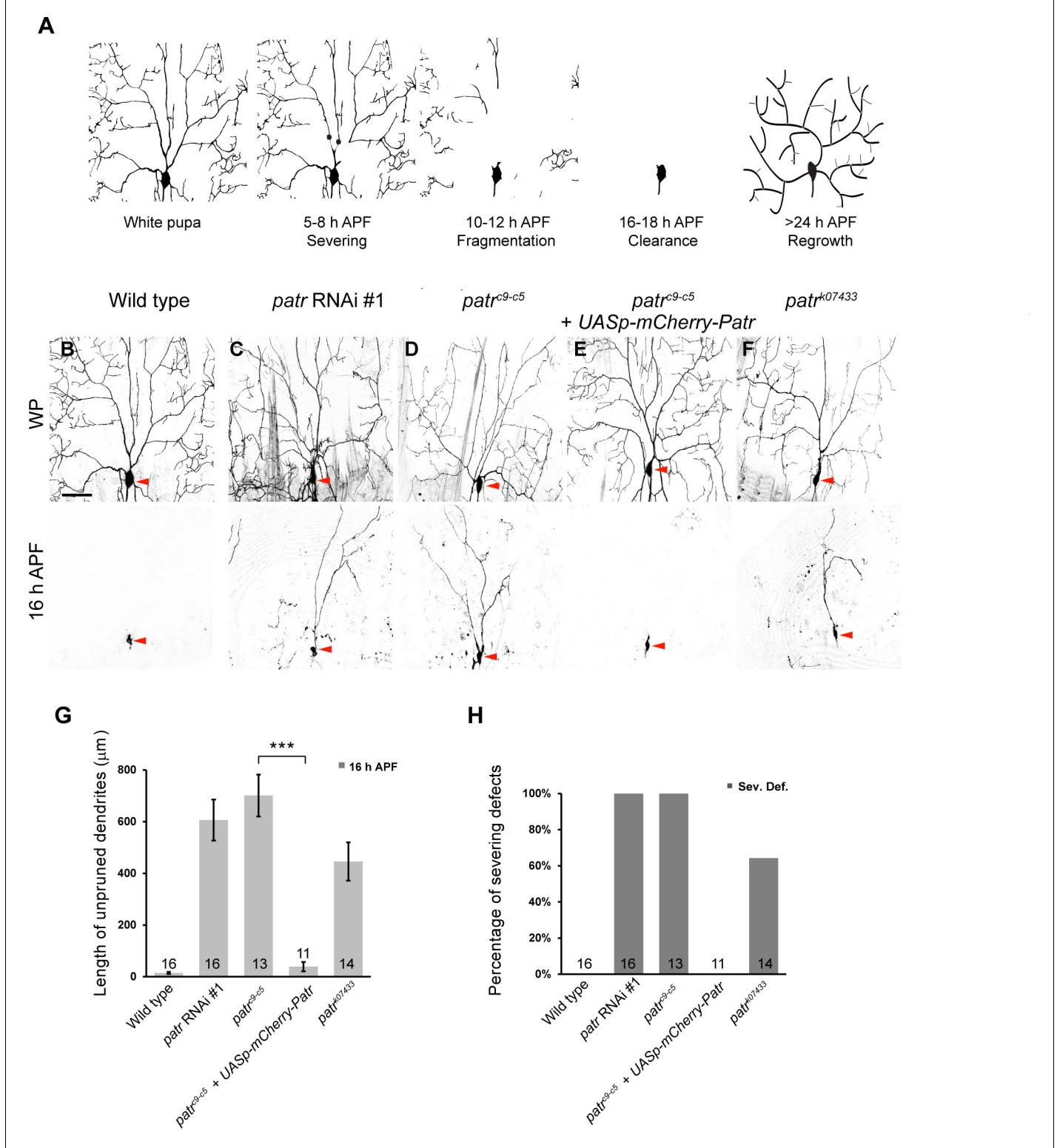

**Figure 1.** Patronin is required for dendrite pruning in ddaC neurons. (**A**) A schematic representation of dendrite pruning in ddaC neurons. (**B–F**) Live confocal images of ddaC neurons expressing mCD8-GFP driven by *ppk-Gal4* at WP and 16 hr APF. While the wild-type neurons pruned all the dendrites (**B**), ddaC neurons overexpressing *patronin* RNAi #1 (**C**), *patronin*^c9-c5^ (**D**) or *patronin*^k07433^ (**F**) MARCM ddaC clones exhibited simple arbors at WP stage and dendrite pruning defects at 16 hr APF. Low-level expression of mCherry-Patronin under the control of the *UASp* promoter fully restored the elaborate arbors at WP and rescued the pruning defects at 16 hr APF in *patronin*^c9-c5^ MARCM ddaC clones (**E**). Red arrowheads point to the ddaC

*Figure 1 continued on next page*

*Figure 1 continued*

somas. (G) Quantification of total length of unpruned ddaC dendrites at 16 hr APF. (H) Quantification of severing defects at 16 hr APF. Scale bar in (B) represents 50 μm. Error bars represent SEM. The number of samples (n) in each group is shown on the bars. ***p<0.001 as assessed by two-tailed Student's T test.

DOI: https://doi.org/10.7554/eLife.39964.002

The following source data and figure supplements are available for figure 1:

**Source data 1.** Extended statistical data as Microsoft Excel spreadsheet.
DOI: https://doi.org/10.7554/eLife.39964.005

**Figure supplement 1.** Patronin is required for dendrite pruning and arborization in sensory neurons.
DOI: https://doi.org/10.7554/eLife.39964.003

**Figure supplement 1—source data 1.** Extended statistical data as Microsoft Excel spreadsheet.
DOI: https://doi.org/10.7554/eLife.39964.004

(*Meng et al., 2008*). Both Patronin and CAMSAP3 can polarize MT arrays via actin-crosslinking proteins (*Ning et al., 2016*; *Nashchekin et al., 2016*; *Khanal et al., 2016*). In hippocampal neurons, CAMSAP2 localizes to the MT minus ends and plays an important role in axon specification and dendrite formation (*Yau et al., 2014*). In *C. elegans* neurons, PTRN-1 is required for synaptic vesicle localization, neurite morphology and axon regeneration (*Chuang et al., 2014*; *Marcette et al., 2014*; *Richardson et al., 2014*). However, to our knowledge, it is completely unknown about a potential role of Patronin in regulating neuronal development including pruning in *Drosophila*. Moreover, whether the minus-end-out MT orientation is critical for dendrite pruning remains poorly understood.

Here, we report that MT minus-end-binding protein Patronin plays a crucial role in dendrite pruning in class IV ddaC neurons. Overexpression of Patronin resembles *patronin* mutants in terms of dendrite pruning phenotypes. The CKK domain is important for Patronin's function in governing dendrite pruning. Moreover, both *patronin* knockdown and overexpression resulted in a drastic decrease of the MT minus-end marker Nod-β-gal and a significant increase of anterograde EB1-GFP comets in proximal ddaC dendrites, indicating that Patronin may stabilize uniform minus-end-out MTs in the dendrites. Interestingly, attenuation of Klp10A, a kinesin-13 MT depolymerase, in *patronin* ddaC neurons significantly restored uniform minus-end-out MTs in dendrites and thereby rescued dendrite pruning defects. Thus, our study demonstrates, for the first time, that the MT minus-end-binding protein Patronin orients uniform minus-end-out MT arrays in dendrites to govern dendrite-specific pruning mainly through antagonizing Klp10A activity in *Drosophila* sensory neurons.

## Results

### Patronin is required for Dendrite pruning and arborization in ddaC neurons

Severing, the first step of dendrite-specific pruning, takes place in the proximal dendrites of C4da or ddaC neurons where MT minus ends predominantly localize (*Rolls et al., 2007*; *Zheng et al., 2008*; *Satoh et al., 2008*). To understand a potential role of MT minus-end-binding proteins in dendrite pruning, we conducted a candidate-based RNA interference (RNAi) screen. Patronin was isolated for its requirement in dendrite pruning. Knockdown of Patronin, via three independent RNAi (#1, v108927; #2, BL36659 and #3, NIG18462Ra-1), resulted in consistent dendrite pruning defects in ddaC neurons at 16 hr APF (*Figure 1C, G and H*, *Figure 1—figure supplement 1A*). In contrast, all the larval dendrites of control neurons were pruned away at the same time point (n = 16, *Figure 1B, G and H*). To further verify this requirement, we generated homozygous MARCM clones of *patronin$^{c9-c5}$*, a *patronin* null mutant (*Nashchekin et al., 2016*). Consistently, mutant ddaC neurons derived from *patronin* RNAi expression (#1, n = 16, *Figure 1C and H*) or *patronin$^{c9-c5}$* mutant clones (n = 13, *Figure 1D and H*) exhibited severe dendrite pruning defects with full penetrance at 16 hr APF. On average, 606 μm and 701 μm of larval dendrites in *patronin* RNAi and *patronin$^{c9-c5}$* ddaC neurons remained attached to their mutant somas, respectively (*Figure 1G*). The dendrites of *patronin* RNAi ddaC neurons were largely removed by 32 hr APF (n = 32; *Figure 1—figure supplement 1B*),

presumably due to extensive apoptosis/migration of the dorsal abdominal epidermis, on which neurons arborize their larval dendrites (*Williams and Truman, 2005*). Moreover, the dendrite severing defects in *patronin*<sup>c9-c5</sup> mutant neurons were completely rescued by low-level expression of mCherry-tagged Patronin protein under the control of the germline *UASp* promoter (*UASp-mCherry-Patronin*) (n = 11, *Figure 1E, G and H*, *Figure 2—figure supplement 1A*). As a control, the expression of *UASp-mCherry-Patronin* did not disturb dendrite pruning in ddaC neurons (n = 16; *Figure 3—figure supplement 1B*). We also made use of a *P-element* insertion line *patronin*<sup>k07433</sup> and observed similar dendrite severing defects in 64% of mutant neurons (n = 14, *Figure 1F and H*). An average of 446 μm dendrites remained in the vicinity of *patronin*<sup>k07433</sup> ddaC clones (*Figure 1G*). Thus, Patronin plays a crucial role in dendrite pruning in ddaC neurons. To our knowledge, this is the first observation indicating that the Patronin family of proteins regulate neuronal remodeling during animal development.

The dendritic complexity of *patronin*<sup>c9-c5</sup> ddaC clones was greatly compromised at the wandering 3<sup>rd</sup> instar larval stage (wL3) (*Figure 1—figure supplement 1C*). The number of their mutant dendritic termini was significantly decreased (n = 5), compared to that of the wild-type controls (n = 5). However, the number of their primary and secondary dendrites in *patronin*<sup>c9-c5</sup> ddaC clones (53.4 ± 4.7) was slightly fewer but not statistically different, compared to that of the control neurons (61.8 ± 2.0). Likewise, at the white prepupal (WP) stage, simplified dendrite arbors formed in *patronin* RNAi (#1, n = 8, *Figure 1C*) or *patronin*<sup>c9-c5</sup> ddaC clones (n = 6, *Figure 1D*), in contrast to more elaborate dendrite arbors observed in those control neurons (n = 19, *Figure 1B*). Mammalian CAMSAP2 was reported to regulate dendrite morphology of cultured hippocampal neurons in vitro (*Yau et al., 2014*). Thus, our in vivo results highlight that Patronin plays a conserved role in regulating dendrite arborization in *Drosophila*.

Moreover, wild-type ddaD/ddaE neurons, like ddaC neurons, pruned all their larval dendrites at 20 hr APF (n = 14, *Figure 1—figure supplement 1D*). *patronin* RNAi-expressing ddaD/ddaE neurons failed to eliminate their dendrites (n = 20, *Figure 1—figure supplement 1D*), similar to *patronin* ddaC neurons. However, *patronin* RNAi-expressing ddaF neurons were still apoptotic (n = 14), similar to wild-type neurons (n = 15, *Figure 1—figure supplement 1E*), suggesting that Patronin is dispensable for neuronal apoptosis during early metamorphosis.

Taken together, Patronin plays a novel and crucial role in dendrite pruning of *Drosophila* sensory neurons.

## Overexpression of patronin causes dendrite pruning defects in ddaC neurons

Given that Patronin acts as a MT minus-end-binding protein to protect MT minus ends and regulate MT behavior, we hypothesized that elevated levels of Patronin might disturb MT dynamics and thereby impair dendrite pruning. We next attempted to investigate the gain-of-function effects of *patronin*. We generated an untagged *patronin* transgene under the control of *UASt* promoter (*UAS-patronin*) and also utilized another previously reported *UASt* transgene (*UAS-GFP-patronin*) (*Derivery et al., 2015*), to induce high-level expression in ddaC neurons (*Figure 2—figure supplement 1A*). Importantly, overexpressing either of *patronin* transgenes, via *Gal4*<sup>4-77</sup> (weak driver) or *ppk-Gal4* (stronger driver), resulted in consistent dendrite pruning defects at 16 hr APF (*Figure 2B and C*), which phenocopied *patronin* mutants (*Figure 1*). Overexpression of Patronin, via *Gal4*<sup>4-77</sup>, led to dendrite severing defects in 70% of ddaC neurons with an average of 511 μm dendrites (n = 34; *Figure 2B, F and G*). When GFP-Patronin was overexpressed at a higher level by two copies of *ppk-Gal4*, vast majority of ddaC neurons exhibited stronger dendrite severing defects with the persistence of 884 μm dendrites (87%, n = 15; *Figure 2C, F and G*), suggesting that overexpressed Patronin behaves as a dominant negative in dendrite pruning.

We also observed that the dendritic complexity was significantly reduced in GFP-Patronin-overexpressing ddaC neurons at wL3 stage (n = 5, *Figure 2—figure supplement 1B*), compared to that in the control neurons (n = 5, *Figure 2—figure supplement 1B*). Likewise, simplified dendrite arbors formed in ddaC neurons overexpressing Patronin (n = 16; *Figure 2B*) or GFP-Patronin (n = 16, *Figure 2C*) at WP stage. To rule out the possibility that the dendrite pruning defects are secondary to the initial morphology defects, we induced the expression of GFP-Patronin at the early 3<sup>rd</sup> instar larval stage (eL3) using the Gene-Switch system. Late-larval induction of GFP-Patronin expression did not affect initial dendrite arborization at WP stage (n = 8; *Figure 2E*). Notably, dendrite severing

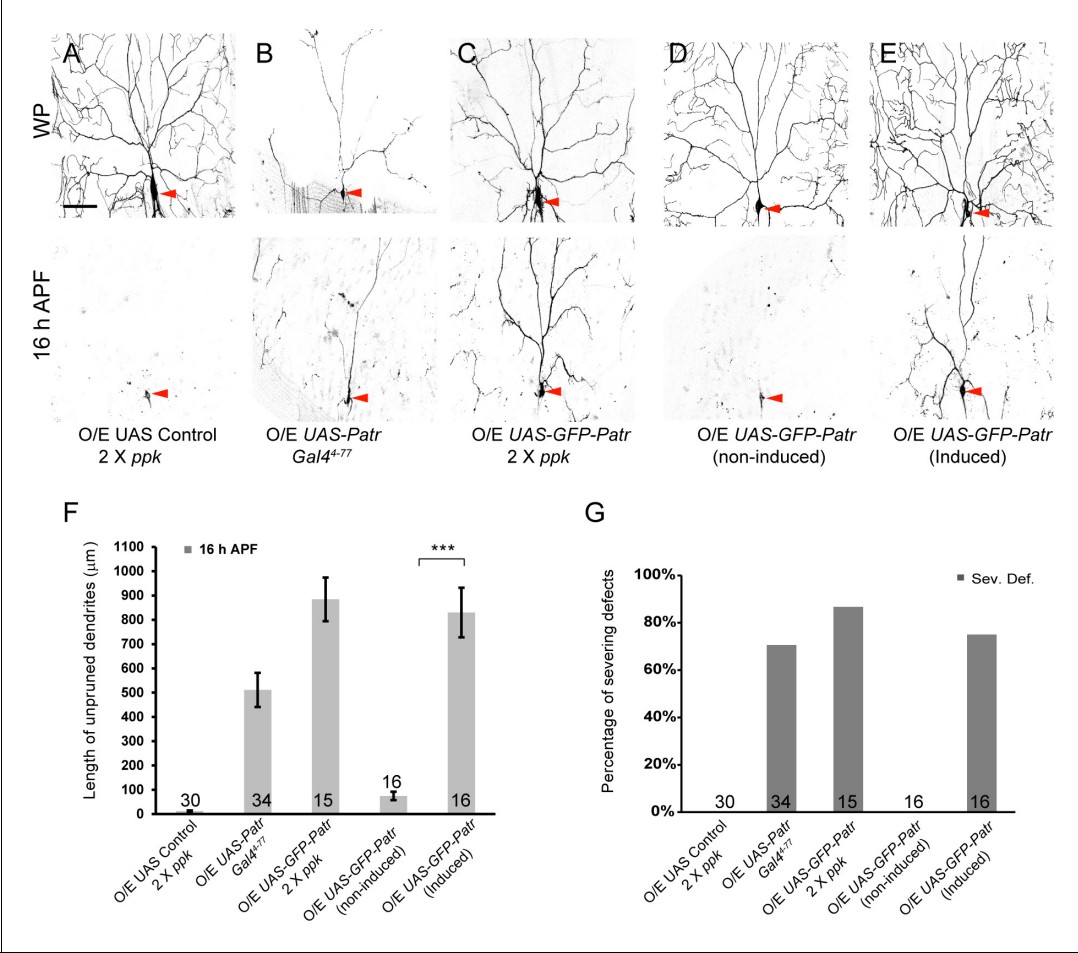

**Figure 2.** Overexpression of Patronin causes dendrite pruning defects in ddaC neurons. (A–E) Live confocal images of ddaC neurons expressing mCD8-GFP driven by *ppk-Gal4, Gal4^{4-77}* or *ppk-CD4-tdGFP* at WP and 16 hr APF. While ddaC neurons overexpressing the *UAS* control construct pruned all the dendrites (A), ddaC neurons overexpressing Patronin (B) driven by *Gal4^{4-77}* or GFP-Patronin (C) driven by two copies of *ppk-Gal4* at a higher level exhibited simple arbors at WP stage and consistent dendrite pruning defects at 16 hr APF. ddaC neurons, in which GFP-Patronin was expressed by *GeneSwitch-Gal4-2295* in RU486-induced conditions (E), exhibited normal arbors at WP stage but severe dendrite pruning defects at 16 hr APF, compared to those in non-induced conditions (D). Red arrowheads point to the ddaC somas. (F) Quantification of total length of unpruned ddaC dendrites at 16 hr APF. (G) Quantification of severing defects at 16 hr APF. Scale bar in (A) represents 50 μm. Error bars represent SEM. The number of samples (n) in each group is shown on the bars. ***p<0.001 as assessed by two-tailed Student's T test.

DOI: https://doi.org/10.7554/eLife.39964.006

The following source data and figure supplements are available for figure 2:

**Source data 1.** Extended statistical data as Microsoft Excel spreadsheet.

DOI: https://doi.org/10.7554/eLife.39964.009

**Figure supplement 1.** Overexpression of Patronin causes dendrite arborization defects in ddaC neurons.

DOI: https://doi.org/10.7554/eLife.39964.007

**Figure supplement 1—source data 1.** Extended statistical data as Microsoft Excel spreadsheet.

DOI: https://doi.org/10.7554/eLife.39964.008

defects were observed at 16 hr APF in 75% of GFP-Patronin-overexpressing ddaC neurons derived from RU486-fed animals (n = 16; *Figure 2E, F and G*), in contrast to no severing defect in non-fed animals (n = 16; *Figure 2D, F and G*).

Thus, Patronin overexpression phenocopies *patronin* mutants in dendrite pruning of sensory neurons.

## The CKK domain is important for patronin to govern dendrite pruning

Patronin contains a CH domain at its amino-terminus, three CC domains at its central region, and a signature CKK domain at its carboxyl-terminus (*Figure 3—figure supplement 1A*) (*Baines et al., 2009*). To dissect functional domains of Patronin that are required for its function in dendrite pruning, we generated a series of *patronin* transgenes (*Figure 3—figure supplement 1A*) under the germline *UASp* promoter, which drives low-level expression in ddaC neurons (*Figure 2—figure supplement 1A*). As controls, we confirmed that low-level expression of these *UASp* transgenes did not result in any dendrite pruning defect in ddaC neurons at 16 hr APF (*Figure 3—figure supplement 1B*). We next examined their abilities to rescue dendrite pruning phenotypes by expressing these Patronin truncates in *patronin*$^{c9-c5}$ mutant MARCM ddaC clones. Notably, the expression of CH-deleted Patronin variant (Patronin$^{\Delta CH}$) was able to almost fully rescue the pruning defects in *patronin*$^{c9-c5}$ ddaC neurons at 16 hr APF (n = 11; *Figure 3B, E and F*), similar to the full-length protein (*Figure 1E, G and H*). However, CKK-deleted Patronin variant (Patronin$^{\Delta CKK}$) failed to rescue the dendrite pruning defects in *patronin*$^{c9-c5}$ ddaC neurons (n = 10; *Figure 3C, E and F*). These data suggest that the CKK domain, rather than the CH domain, is required for Patronin's function in dendrite pruning. To further examine whether the CKK domain alone is able to substitute for Patronin's function in dendrite pruning, we expressed the CKK domain in *patronin*$^{c9-c5}$ ddaC MARCM clones. Interestingly, the expression of the CKK domain significantly rescued *patronin*$^{c9-c5}$-associated dendrite pruning defects (n = 16; *Figure 3D, E and F*). Thus, the CKK domain is important for Patronin's function during dendrite pruning.

To further investigate gain-of-function effects of the Patronin variants, we generated a second set of *patronin* transgenes under the *UASt* promoter for high-level expression in ddaC neurons. Overexpression of Patronin$^{\Delta CH}$ variant led to prominent dendrite severing defects in ddaC neurons at 16 hr APF (59%, n = 27, *Figure 3H, K and L*), similar to the full-length Patronin (69%, n = 16; *Figure 3G, K and L*). In contrast, the expression of Patronin$^{\Delta CKK}$ variant did not affect normal dendrite pruning at 16 hr APF (n = 16; *Figure 3I, K and L*), suggesting that the CKK domain of Patronin is important for its gain-of-function effect on dendrite pruning. Interestingly, overexpression of CKK domain (n = 16; *Figure 3J, K and L*), CH domain (n = 25; *Figure 3—figure supplement 1A*) or CC1-3 domains (n = 17; *Figure 3—figure supplement 1A*) had no effect on dendrite pruning of ddaC neurons, suggesting that endogenous Patronin protein may function properly to regulate dendrite pruning even with ectopic expression of these individual domains.

Collectively, both rescue and overexpression results strengthen the conclusion that the CKK domain is indispensable for Patronin to exert its functions in dendrite pruning.

## Patronin is required for proper distribution of dendritic and axonal MT markers

We next attempted to understand the mechanisms whereby Patronin regulates dendrite-specific pruning in ddaC sensory neurons. Since Patronin binds and protects MT minus ends in *Drosophila* S2 cells (*Goodwin and Vale, 2010*), we hypothesized that Patronin may stabilize MT minus ends and govern MT minus-end distribution in ddaC sensory neurons. The chimeric protein Nod-β-gal, in which the Nod motor domain is fused to the coiled-coil domain region of the conventional kinesin1 (Kin1) and the β-galactosidase (β-gal) tag, localizes to the MT minus ends and is commonly used as a marker of MT minus ends in *Drosophila* (*Clark et al., 1997*). In *Drosophila* neurons including da neurons, this Nod-β-gal chimera localizes specifically in dendrites but not in axons (*Rolls et al., 2007*; *Zheng et al., 2008*; *Satoh et al., 2008*). In wild-type ddaC neurons, Nod-β-gal was predominantly enriched in dendrites but not in axons (n = 6, *Figure 4A, D and E*). Remarkably, in all *patronin* RNAi ddaC neurons, Nod-β-gal was no longer enriched in the dendritic arbors and instead highly concentrated in the soma (100%, n = 25, *Figure 4B and D*). Likewise, all *patronin*$^{c9-c5}$ ddaC clones also exhibited robust accumulation of Nod-β-gal in the soma concomitantly with its reduced levels in the distal dendrites (100%, n = 11; *Figure 4—figure supplement 1A*, *Figure 4D and E,*). To quantify the alterations of dendritic Nod-β-gal distribution, we measured its intensity in the dendrites which were 40 μm away from the soma. In the dendrites of *patronin* RNAi or *patronin*$^{c9-c5}$ neurons, Nod-β-gal levels were drastically reduced to 12% and 11% of those in the control neurons, respectively (*Figure 4E*). Reduced levels of dendritic Nod-β-gal signals may reflect a decrease in the number of minus-end-out MTs in the dendrites of *patronin* mutant neurons. Strikingly, overexpression of GFP-

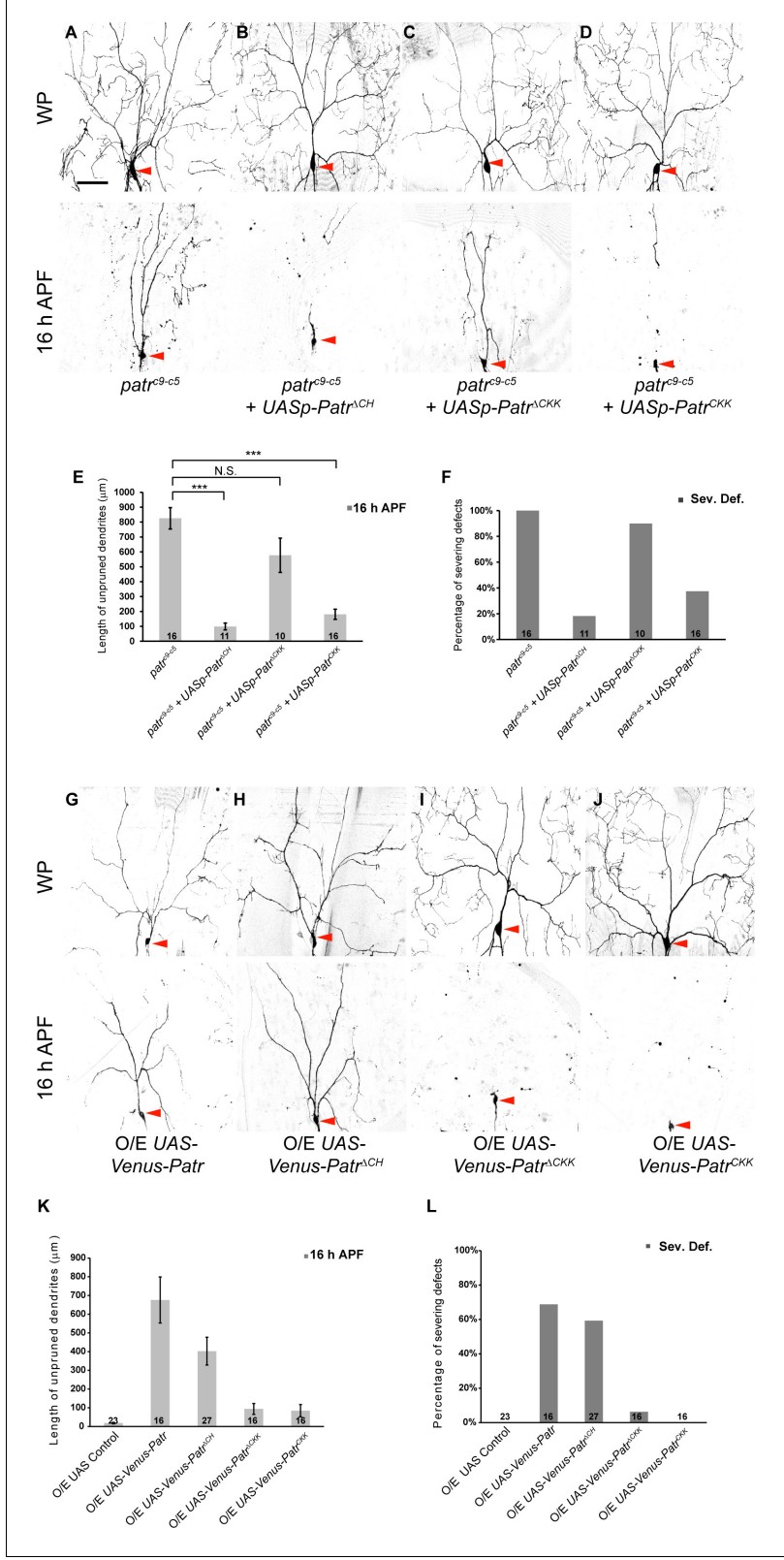

**Figure 3.** The CKK domain is important for Patronin to govern dendrite pruning. (**A–D, G–J**) Live confocal images of ddaC neurons expressing mCD8-GFP driven by *ppk-Gal4* or *Gal4^4-77* at WP and 16 hr APF. Low-level expressions of Patronin^ΔCH (**B**) and Patronin^CKK (**D**), but not Patronin^ΔCKK (**C**), strongly rescued the dendrite arborization defects at WP stage and the pruning defects at 16 hr APF in *patronin^c9-c5* (**A**) MARCM ddaC clones.
*Figure 3 continued on next page*

*Figure 3 continued*

(G–J) ddaC neurons overexpressing Venus-Patronin (G) or Venus-Patronin$^{\Delta CH}$ (H) by *Gal4$^{4-77}$* exhibited simple arbors at WP stage and the dendrite pruning defects at 16 hr APF. However, overexpression of Venus-Patronin$^{\Delta CKK}$ (I) or Venus-Patronin$^{CKK}$ (J) had negligible effect on dendrite arborization and pruning. (E, K) Quantification of total length of unpruned ddaC dendrites at 16 hr APF. (F, L) Quantification of severing defects at 16 hr APF. Scale bar in (A) represents 50 μm. Error bars represent SEM. The number of samples (n) in each group is shown on the bars. N.S., not significant; ***p<0.001 as assessed by one-way ANOVA test.
DOI: https://doi.org/10.7554/eLife.39964.010

The following source data and figure supplements are available for figure 3:

**Source data 1.** Extended statistical data as Microsoft Excel spreadsheet.
DOI: https://doi.org/10.7554/eLife.39964.013
**Figure supplement 1.** Structure-function analysis of the Patronin protein.
DOI: https://doi.org/10.7554/eLife.39964.011
**Figure supplement 1—source data 1.** Extended statistical data as Microsoft Excel spreadsheet.
DOI: https://doi.org/10.7554/eLife.39964.012

Patronin also resulted in robust accumulation of Nod-β-gal signals in the soma and reduced dendritic distribution in 67% of ddaC neurons (n = 18, *Figure 4C, D and E*), resembling the *patronin* mutant phenotypes. No mis-localization of Nod-β-gal signals was observed in the axons of *patronin* mutant or GFP-Patronin-overexpressing ddaC neurons (*Figure 4B and C*). These results imply that Patronin might stabilize and distribute MT minus ends into the dendrites of ddaC neurons. Collectively, Patronin regulates dendritic localization of the MT minus-end marker Nod-β-gal in a dose-sensitive manner.

We next examined the distribution of the axon-specific marker Kin-β-gal, which consists of the Kin1 motor and coiled-coil domains fused with the β-gal tag and is often used as a marker of MT plus ends (*Clark et al., 1997*). Kin-β-gal was localized exclusively to the axons, but not to the dendrites in wild-type ddaC neurons (n = 9; *Figure 4F and I*) (*Zheng et al., 2008*). Interestingly, Kin-β-gal often mis-localized to the dendrites as punctate structures in *patronin* RNAi (89%, n = 19, *Figure 4G and I*) or GFP-Patronin-overexpressing ddaC neurons (45%, n = 20, *Figure 4H and I*), implying an increase of plus-end-out MTs in the dendrites. Given that Patronin localizes at MT minus ends in S2 cells (*Goodwin and Vale, 2010*), we next examined its subcellular localization in ddaC neurons using the anti-Patronin antibody. Patronin localized uniformly to dendrites, axons and soma at larval and WP stages (n = 14 and 14, respectively, *Figure 4—figure supplement 1B*).

Taken together, Patronin is crucial for proper distribution of dendrite or axon-specific MT markers in ddaC neurons.

## Patronin is required for the minus-end-out orientation of dendritic MTs in ddaC neurons

Given that Patronin regulates proper distribution of dendritic or axonal MT markers in ddaC neurons, we next assessed whether dendritic MT minus-end-out orientation depends on Patronin. To this end, we took advantage of the MT plus-end marker EB1-GFP. EB1 is a core plus-end-binding protein that associates with the growing plus ends of MTs during growth and dissociates during shrinkage (*Vaughan, 2005*). GFP-fused EB1 was reported to track the growing plus ends of MTs in neurons (*Stepanova et al., 2003*; *Rolls et al., 2007*). To assess MT orientation in ddaC dendrites, we expressed low-level EB1-GFP via the weak driver *Gal4$^{4-77}$* to examine the direction of EB1-GFP comet movement in major dendrites at 96 hr after egg laying (AEL) (*Rolls et al., 2007*). We first knocked down *patronin* by using two independent RNAi lines and compared them with the control RNAi line. In the proximal dendrites, 99% of EB1-GFP comets moved predominantly toward the soma (retrograde) in the control RNAi-expressing ddaC neurons (n = 30, *Figure 5A and C*). These retrograde EB1-GFP events indicate a predominant minus-end-out orientation of dendritic MTs in ddaC neurons (*Stone et al., 2008*; *Ori-McKenney et al., 2012*). In contrast, in the control ddaC axons, EB1-GFP comets primarily moved away from the soma (anterograde, 96%, n = 12, *Figure 5—figure supplement 1C*), suggesting a plus-end-out MT orientation in axons. On average, the percentage of anterograde EB1-GFP comets was increased to 48% and 19% in these *patronin* RNAi ddaC neurons, respectively (#1, n = 29 neurons, *Figure 5B–C*; #2, n = 12, *Figure 5—figure*

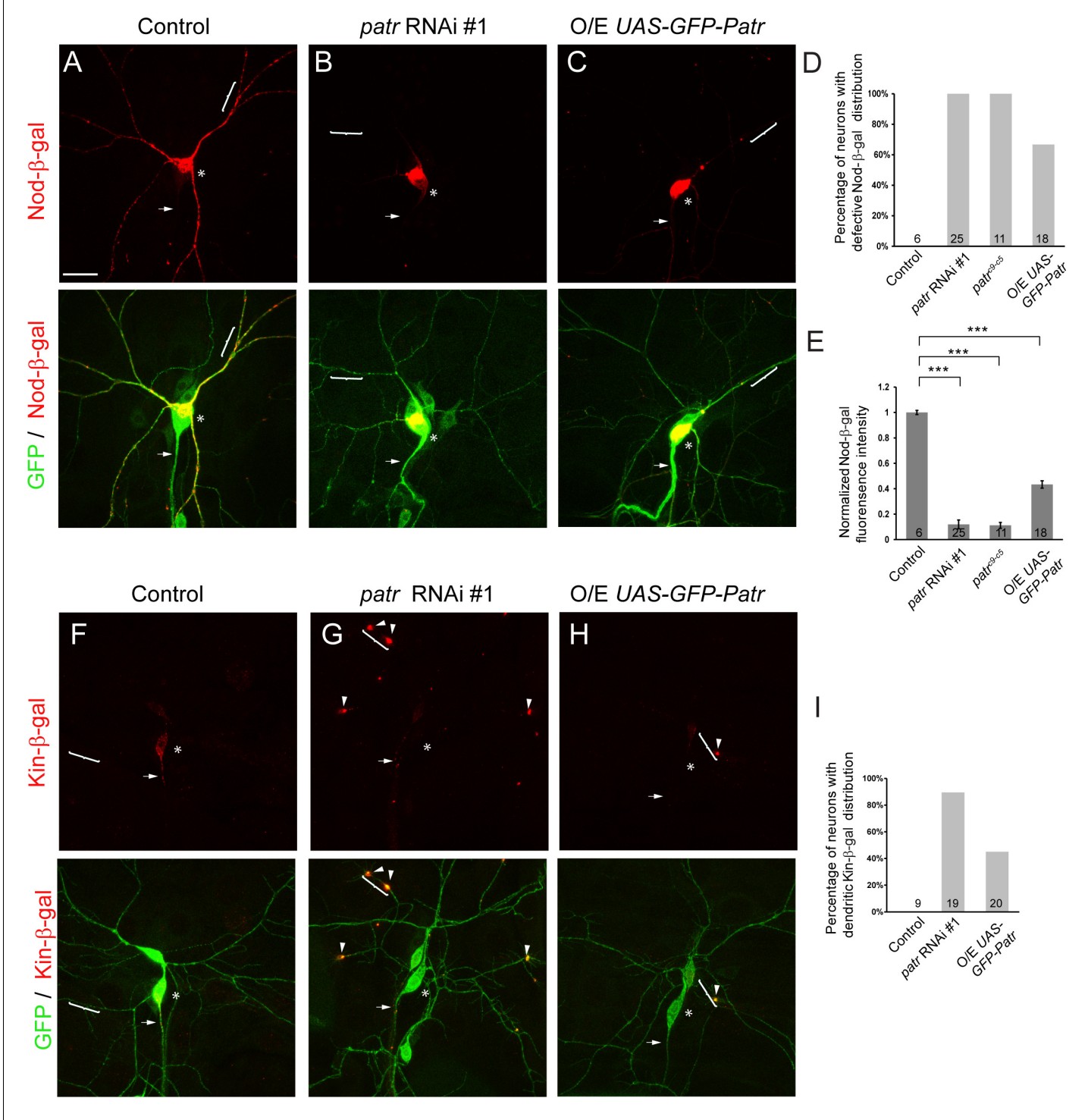

**Figure 4.** Patronin is required for proper distribution of dendritic and axonal MT markers. (A–C, F–H) Confocal images of ddaC neurons expressing mCD8-GFP, Nod-β-gal or Kin-β-gal and immunostained for β-galactosidase at wL3 stage. ddaC somas are marked by asterisks, axons by arrows and dendrites by curly brackets. Nod-β-gal levels were reduced in the dendrites but enriched in the somas in *patronin* RNAi #1 (B) or *GFP-Patronin*-overexpressing (C) ddaC neurons, compared to the control neurons (A). (D) Quantification of the percentage of neurons with defective Nod-β-gal distribution. (E) Quantification of normalized Nod-β-gal intensity in the dendrites. Moreover, Kin-β-gal mis-localized to the dendrites in both *patronin* RNAi #1 (G) or GFP-Patronin-overexpressing (H) ddaC neurons, compared to the control neurons (F). Arrowheads point to ectopic Kin-β-gal aggregates in the dendrites. (I) Quantification of the percentage of neurons with dendritic Kin-β-gal distribution. Scale bar in (A) represents 20 μm. Error bars represent SEM. The number of samples (n) in each group is shown on the bars. ***p<0.001 as assessed by one-way ANOVA test.

*Figure 4 continued on next page*

*Figure 4 continued*

DOI: https://doi.org/10.7554/eLife.39964.014

The following source data and figure supplement are available for figure 4:

**Source data 1.** Extended statistical data as Microsoft Excel spreadsheet.

DOI: https://doi.org/10.7554/eLife.39964.016

**Figure supplement 1.** Patronin is required for proper distribution of the MT marker Nod-β-gal in ddaC dendrites.

DOI: https://doi.org/10.7554/eLife.39964.015

*supplement 1A*), compared to that in the control neurons (n = 30; *Figure 5C*). Notably, the direction of EB1-GFP comets was drastically reversed in 33% of dendrite branches of *patronin* RNAi #1 ddaC neurons, respectively (n = 43 branches; *Figure 5M*). In these mutant dendrite branches, over 80% of EB1-GFP comets moved away from the soma, suggesting a predominant plus-end-out MT pattern (*Figure 5M*). We next quantified the number of EB1-GFP events, which reflects overall MT density and levels of MT nucleation in both cultured cells (*Piehl et al., 2004*) and *Drosophila* neurons (*Chen et al., 2014*). The average numbers of EB1-GFP comets in *patronin* RNAi #1 expressing ddaC dendrites were not statistically different from that in the control (*Figure 5D*). *patronin* RNAi knockdown (#1) also led to normal EB1-GFP track length (*Figure 5E*) but slightly higher growth speed (*Figure 5F*).

The intensity of overall MT levels was significantly reduced in the dendrites of *patronin* RNAi ddaC neurons (#1, n = 16), compared with the controls (n = 16; *Figure 5—figure supplement 1B*), as assessed by the antibody 22C10 against Futsch. Moreover, we have also examined EB1-GFP movement in the axons of *patronin* RNAi ddaC neurons. Compared with 4% of retrograde EB1-GFP comets (predominantly plus-end-out) in the axons of wild-type neurons (n = 12), *patronin* knockdown led to 41% of retrograde EB1-GFP comets in the axons (n = 17; *Figure 5—figure supplement 1C*), indicative of a severe defect in the plus-end-out orientation of their axonal MTs upon *patronin* knockdown. Thus, *patronin* plays an important role in governing both dendritic and axonal MT polarity in ddaC neurons.

Similar to *patronin* RNAi neurons, Patronin overexpression also led to a significant increase in anterograde EB1-GFP comets in the ddaC dendrites (10%, n = 15; *Figure 5H and I*), compared to the control neurons (0%, n = 12; *Figure 5G and I*). The number of EB1-GFP comets in Patronin-overexpressing ddaC dendrites remained similar to that of the controls (*Figure 5I*). Patronin overexpression also led to an increase in the number of EB1-GFP comets (*Figure 5J*) but did not affect normal EB1-GFP track length (*Figure 5K*) and growth speed (*Figure 5L*).

Thus, Patronin plays an important role in orienting minus-end-out MTs in ddaC dendrites.

## Attenuation of Klp10A, a kinesin-13 MT depolymerase, suppresses the *patronin* phenotype in MT orientation in ddaC dendrites

A previous study revealed that Patronin stabilizes MT minus ends and protects them against kinesin-13-mediated MT depolymerization in interphase and mitotic S2 cells (*Goodwin and Vale, 2010*). We next investigated whether mis-orientation of minus-end-out MTs in *patronin* dendrites is caused by excessive kinesin-13-dependent depolymerization activity. To this end, we double knocked down the *Drosophila* kinesin-13 Klp10A with Patronin and subsequently examined EB1-GFP movements in the dendrites. In 26% of dendrite branches of ddaC neurons expressing both *patronin* RNAi (#1) and control RNAi constructs (*patronin +control* RNAi), EB1-GFP comets predominantly moved away from the soma (n = 23 branches; *Figure 6—figure supplement 1A*), indicating a nearly uniform plus-end-out MT pattern. On average, 56% of EB1-GFP comets moved anterograde in these dendrites of *patronin +control* RNAi neurons (n = 12, *Figure 6C*), similar to *patronin* RNAi-expressing ddaC (*Figure 5C*). Double knockdown of *patronin* and *klp10A* (*patronin +klp10A*) completely restored the retrograde movement of EB1-GFP comets (n = 11; *Figure 6B and C*). Moreover, all dendrite branches exhibited a uniform minus-end-out MT pattern in *patronin +klp10A* RNAi neurons (n = 27 branches; *Figure 6—figure supplement 1A*), identical to the control neurons (*Figure 5M*). Moreover, further knockdown of *klp10A* resulted in no alteration in the number, track length and movement speed of EB1-GFP comets in *patronin* RNAi dendrites (*Figure 6D–F*). As a control, single RNAi knockdown of *klp10A* did not alter the direction of EB1-GFP movements in ddaC dendrites

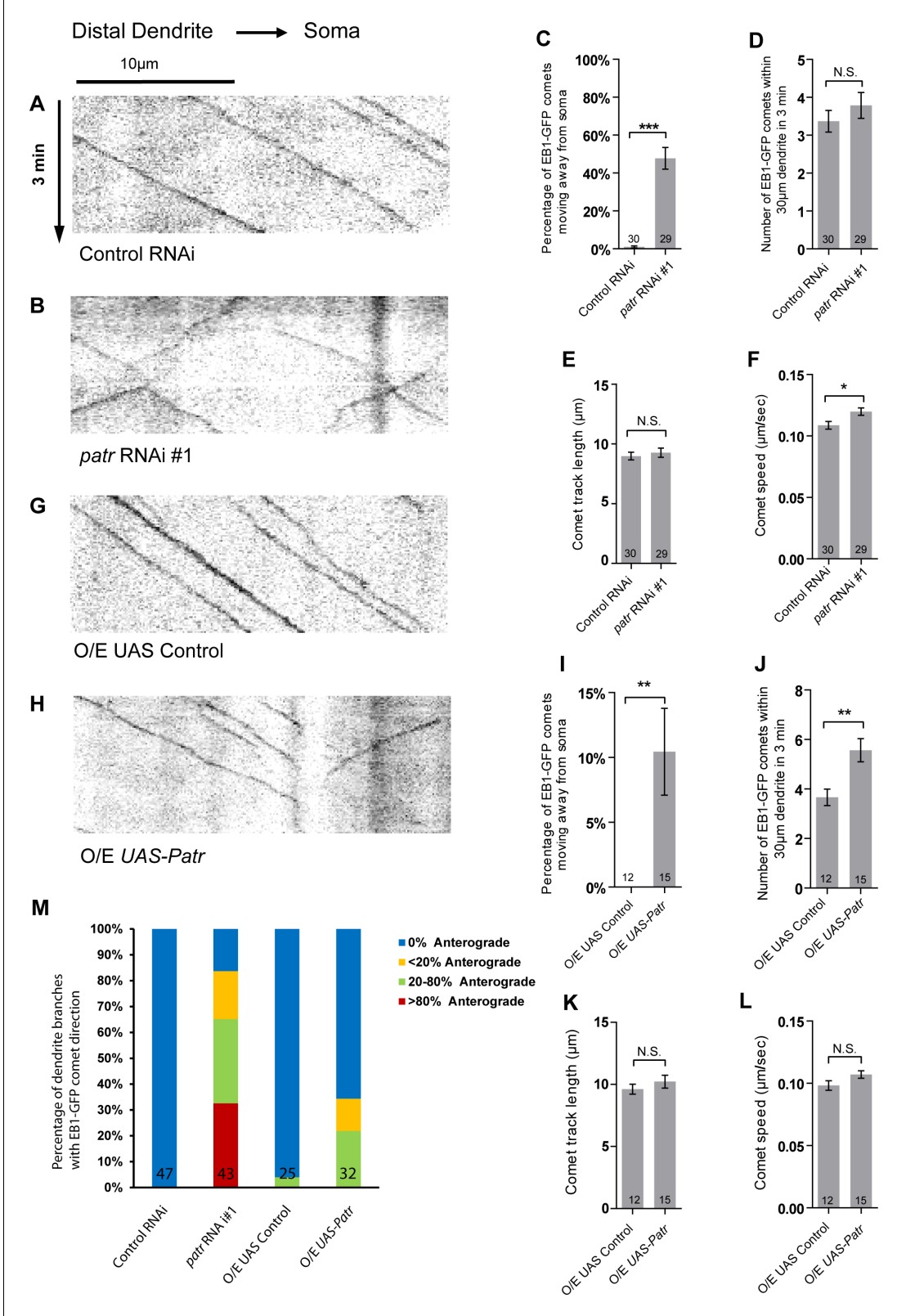

**Figure 5.** Patronin is required for the minus-end-out orientation of dendritic MTs in ddaC neurons. (**A–B, G–H**) Representative kymographs of EB1-GFP comets in control or mutant ddaC dendrites. In control RNAi (**A**) or *UAS* control (**G**) ddaC neurons, dendritic EB1-GFP comets predominantly moved retrogradely towards the somas. However, in *patronin* RNAi #1 (**B**) or *Patronin*-overexpressing (**H**) ddaC neurons, dendritic EB1-GFP comets moved bidirectionally (anterogradely and retrogradely). Horizontal arrow indicates the direction towards the soma. Scale bar in (**A**) represents 10 µm, and each

*Figure 5 continued on next page*

*Figure 5 continued*

movie was taken for 3 min. EB1-GFP was expressed by *Gal4^4-77*. (**C, I**) Quantification of percentage of dendritic EB1-GFP comets moving away from soma (anterogradely). (**D, J**) Quantification of average numbers of EB1-GFP comets along 30 µm of dendrites within 3 min. (**E, K**) Quantification of average track length of EB1-GFP comets along 30 µm of dendrites within 3 min. (**F, L**) Quantification of average speed of EB1-GFP comets along 30 µm of dendrites within 3 min. (**M**) Quantification of the percentage of dendrite branches showing different levels of anterograde EB1-GFP comets. The number of dendrite branches (**n**) examined in each group is shown on the bars. Error bars represent SEM. N.S., not significant; *p<0.05; **p<0.01; ***p<0.001 as assessed by two-tailed Student's T test.

DOI: https://doi.org/10.7554/eLife.39964.017

The following source data and figure supplements are available for figure 5:

**Source data 1.** Extended statistical data as Microsoft Excel spreadsheet.
DOI: https://doi.org/10.7554/eLife.39964.020
**Figure supplement 1.** Patronin is required for dendritic and axonal MT orientations in ddaC neurons.
DOI: https://doi.org/10.7554/eLife.39964.018
**Figure supplement 1—source data 1.** Extended statistical data as Microsoft Excel spreadsheet.
DOI: https://doi.org/10.7554/eLife.39964.019

(n = 12; *Figure 6—figure supplement 1B*). Thus, these results indicate that MT mis-orientation in *patronin* mutant dendrites is caused by excessive Klp10A activity.

We further explored whether aberrant distribution of MT markers, Nod-β-gal and Kin-β-gal, is attributable to excessive Klp10A activity. While Nod-β-gal was predominantly enriched in the soma and strongly reduced in the dendrites in *patronin +control* RNAi ddaC neurons (100%, n = 10, *Figure 6H, J and K*), dendritic distribution of Nod-β-gal was significantly restored in *patronin +klp10A* RNAi mutant ddaC neurons (n = 12, *Figure 6I, J and K*) to an extent similar to that in wild-type controls (n = 13, *Figure 6G, J and K*). As a control, single RNAi knockdown of *klp10A,* via two independent constructs, did not alter the Nod-β-gal distribution in ddaC dendrites (n = 17 and 17, respectively, *Figure 6—figure supplement 1C*). Likewise, mis-localization of the axonal marker Kin-β-gal to the dendrites in *patronin* RNAi neurons was rescued by further knockdown of *klp10A* (22%, n = 18, *Figure 6N and O*), compared to the control RNAi construct (85%, n = 20, *Figure 6M and O*). Thus, attenuation of Klp10A activity restores normal MT orientation in the dendrites of *patronin* mutant neurons.

These results suggest that Patronin acts against Klp10A-dependent depolymerization to maintain uniform minus-end-out MT orientation in the dendrites of ddaC neurons.

## Patronin promotes dendrite pruning by orienting uniform minus-end-out MT arrays in dendrites

Given that attenuation of Klp10A is sufficient to restore uniform minus-end-out orientation of dendritic MTs in *patronin* mutant neurons, we further assessed whether such a restoration of MT orientation also results in a rescue of dendrite pruning defects in mutant neurons. 84% of *patronin +control* RNAi co-overexpressing ddaC neurons displayed dendrite severing defects and retained an average of 593 µm larval dendrite at 16 hr APF (n = 32, *Figure 7A, D and E*). We then co-expressed the *patronin* RNAi #1 line with either of two *klp10A* RNAi lines, v41534 (#1) and BL33963 (#2). Importantly, the pruning defects associated with *patronin* RNAi ddaC neurons were almost completely rescued by further knockdown of *klp10A* (*Figure 7B and C*). On average, only 73 µm and 71 µm of dendrites were present at 16 hr APF in *patronin +klp10A* #1 or #2 double RNAi ddaC neurons (n = 20, *Figure 7B and D* and n = 32, *Figure 7C and D*, respectively). Moreover, the penetrance of severing defects in *patronin +klp10A* #1 or *patronin +klp10A* #2 double RNAi ddaC neurons (*Figure 7E*) was drastically reduced to 15% and 19%, respectively, in contrast to 84% in the *patronin +control* RNAi controls (*Figure 7E*). As controls, knockdown of either *klp10A* RNAi #1 or #2 did not cause any dendrite pruning defects in 16-h-APF ddaC neurons (*Figure 7—figure supplement 1A*). Together with the restoration of dendritic MT orientation in *patronin* mutant neurons by *klp10A* knockdown, these results suggest that the dendrite pruning defects in *patronin* mutant ddaC neurons is mainly attributable to mis-orientation of MT minus ends in the dendrites.

Previous studies reported a direct interaction between mammalian Patronin (CAMSAPs) and katanin (*Jiang et al., 2014; Jiang et al., 2018*). We therefore examined the potential genetic interaction between *patronin* and *kat-60* in dendrite pruning. Knockdown of *kat-60*, via two distinct RNAi lines,

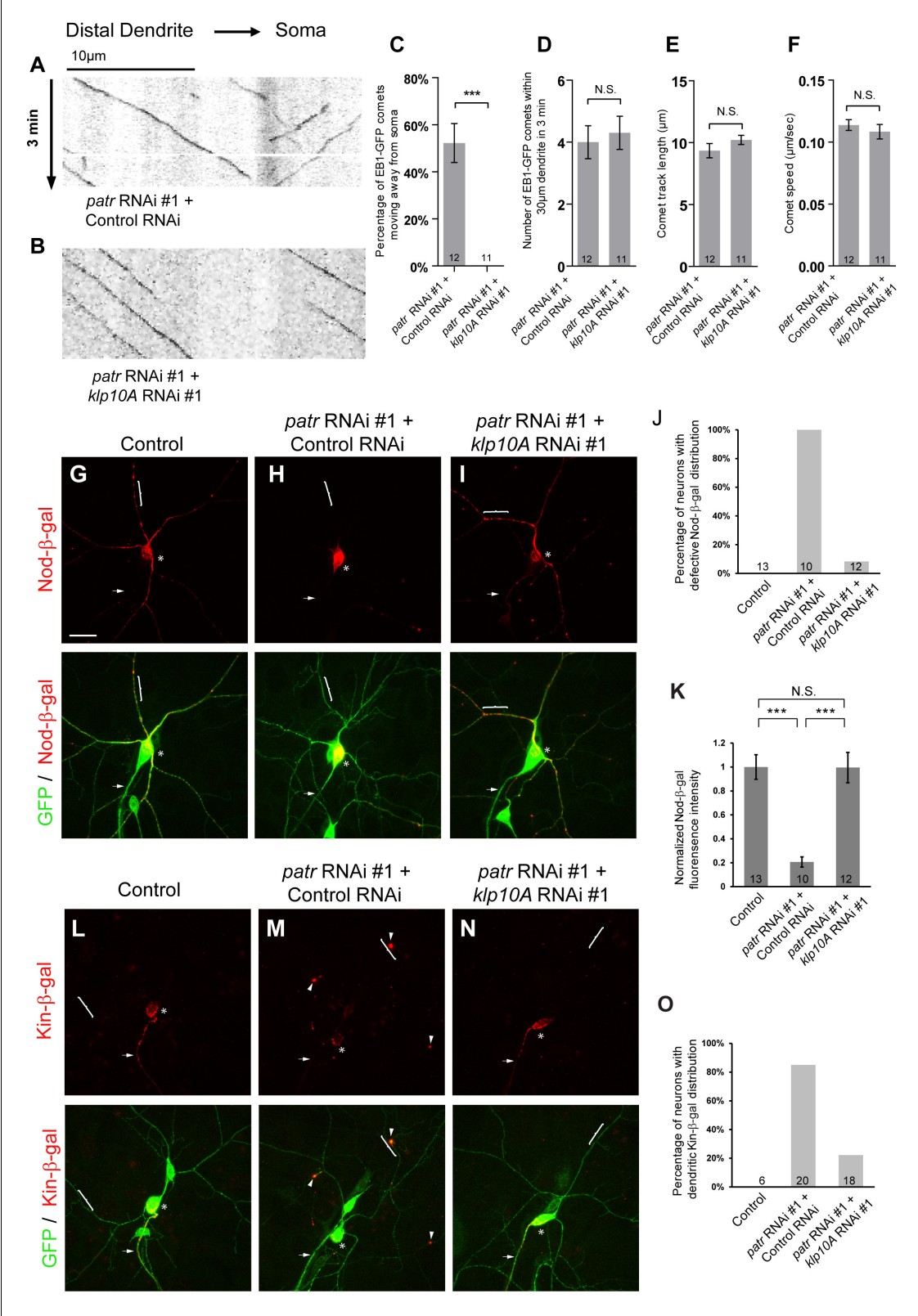

**Figure 6.** Attenuation of Klp10A, a kinesin-13 MT depolymerase, suppresses the *patronin* phenotypes in MT orientation in ddaC dendrites. (**A, B**) Representative kymographs of EB1-GFP comets in control or mutant ddaC dendrites. In *patronin* RNAi #1 and control RNAi co-expressing ddaC neurons (**A**), dendritic EB1-GFP comets moved bidirectionally (anterogradely and retrogradely). However, knockdown of *klp10A* via RNAi #1 completely restored the retrograde movement of EB1-GFP comets in the *patronin* RNAi #1 ddaC dendrites (**B**). Horizontal arrow indicates the direction towards the
*Figure 6 continued on next page*

*Figure 6 continued*

somas. Scale bar in (**A**) represents 10 µm, and each movie was taken for 3 min. EB1-GFP was expressed by *Gal4^4-77^*. (**C**) Quantification of percentage of dendritic EB1-GFP comets moving away from soma (anterogradely). (**D**) Quantification of average numbers of EB1-GFP comets along 30 µm of dendrites within 3 min. (**E**) Quantification of average track length of EB1-GFP comets along 30 µm of dendrites within 3 min. (**F**) Quantification of average speed of EB1-GFP comets. (**G–I, L–N**) Confocal images of ddaC neurons expressing mCD8-GFP, Nod-β-gal or Kin-β-gal and immunostained for β-galactosidase at wL3 stage. ddaC somas are marked by asterisks, axons by arrows and dendrites by curly brackets. Nod-β-gal levels were reduced in the dendrites but enriched in the somas of ddaC neurons co-overexpressing *patronin* RNAi #1 and control RNAi (**H**), compared to the control ddaC neurons (**G**). However, knockdown of *klp10A* (RNAi #1) almost completely restored dendritic distribution of Nod-β-gal in *patronin* RNAi #1 ddaC neurons (**I**). (**J**) Quantification of the percentage of neurons with defective Nod-β-gal distribution. (**K**) Quantification of normalized Nod-β-gal intensity in the dendrites. Moreover, Kin-β-gal mis-localization defects in *patronin* RNAi #1 ddaC neurons were drastically rescued by knockdown of *klp10A* (RNAi #1), compared to the *patronin*, control RNAi neurons (**M**). (**O**) Quantification of the percentage of neurons with dendritic Kin-β-gal distribution. Scale bar in (**G**) represents 20 µm. Error bars represent SEM. The number of samples (n) in each group is shown on the bars. N.S., not significant; ***p<0.001 as assessed by one-way ANOVA test or two-tailed Student's T test.

DOI: https://doi.org/10.7554/eLife.39964.021

The following source data and figure supplements are available for figure 6:

**Source data 1.** Extended statistical data as Microsoft Excel spreadsheet.

DOI: https://doi.org/10.7554/eLife.39964.024

**Figure supplement 1.** Attenuation of Klp10A, a kinesin-13 MT depolymerase, suppresses the *patronin* phenotype in MT orientation in ddaC dendrites.

DOI: https://doi.org/10.7554/eLife.39964.022

**Figure supplement 1—source data 1.** Extended statistical data as Microsoft Excel spreadsheet.

DOI: https://doi.org/10.7554/eLife.39964.023

did not affect normal dendrite pruning in ddaC neurons (*Figure 7—figure supplement 1B*). Knockdown of *kat-60* did not enhance or suppress the pruning defects in *patronin* RNAi ddaC neurons (*Figure 7—figure supplement 1C*). Likewise, although Kat-60L1 and Tau were reported to be involved in dendrite pruning (*Lee et al., 2009*; *Herzmann et al., 2017*), double knockdown of *kat-60L1/tau* and *patronin* exhibited no significant enhancement or suppression in the dendrite pruning defects (*Figure 7—figure supplement 1D–E*). These data suggest no genetic interaction between *patronin* and *kat-60/kat-60L1/tau*.

To address whether Patronin overexpression, like *patronin* knockdown, also affects dendrite pruning through excessive Klp10A activity, we knocked down *klp10A* function in Patronin-overexpressing ddaC neurons. Knockdown of *klp10A*, via two independent RNAi lines, significantly suppressed the dendrite pruning defects in Patronin-overexpressing ddaC neurons (n = 17 and 24; *Figure 7G–J*). Moreover, knockdown of *klp10A* also significantly suppressed the MT orientation defect in the dendrites of Patronin-overexpressing ddaC neurons (n = 14; *Figure 7L–N*), compared to the control neurons (n = 13, *Figure 7K and M–N*). Therefore, these data further confirm that Patronin overexpression, like *patronin* knockdown, affects dendrite pruning and dendritic MT orientation at least partially by upregulation of Klp10A function.

Taken together, Patronin regulates dendritic minus-end-out MT orientation and dendrite pruning at least partially by antagonizing Klp10A function.

## Klp10A overexpression phenocopies *patronin* knockdown in dendrite pruning and dendritic MT orientation

To further confirm whether the dendrite pruning defects in *patronin* mutant neurons is due to increased Klp10A activity, we overexpressed Klp10A in ddaC neurons via two independent transgenes, *UAS-klp10A* and *UAS-GFP-klp10A*. Indeed, Klp10A overexpression alone is sufficient to inhibit dendrite pruning in ddaC neurons (*Figure 8B–E*), phenocopying *patronin* mutants (*Figure 1*). Overexpression of either GFP-Klp10A (n = 16, *Figure 8B, D and E*) or Klp10A (n = 16, *Figure 8C, D and E*) resulted in consistent severing pruning defects in 88% and 94% of neurons with the persistence of 662 µm and 722 µm dendrites present at 16 hr APF, respectively. Similar to *patronin* mutants, overexpression of GFP-Klp10A (n = 8; *Figure 8B*) or Klp10A (n = 8; *Figure 8C*) also resulted in simplified dendrite arbors at WP stage, suggesting that elevated Klp10A impairs initial dendrite arborization during growth. Moreover, overexpression of Klp10A also resulted in reduced levels of Nod-β-gal in the dendrites (n = 17; *Figure 8—figure supplement 1A*) and mis-localization of Kin-β-gal to the dendrites (n = 20; *Figure 8—figure supplement 1B*). Consistently, Klp10A

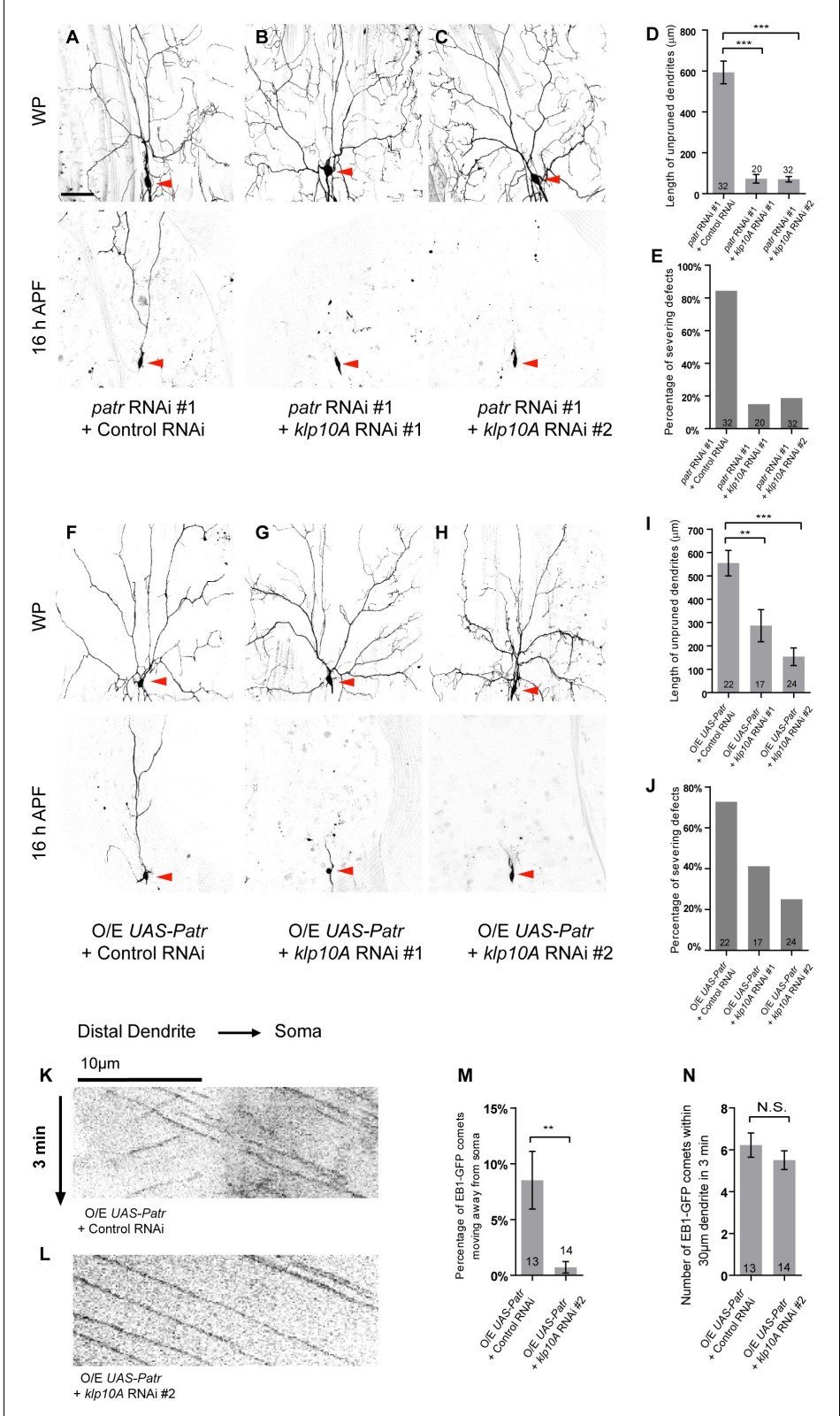

**Figure 7.** Patronin promotes dendrite pruning by orienting uniform minus-end-out MT arrays in dendrites. (**A–C, F–H**) Live confocal images of ddaC neurons expressing mCD8-GFP driven by *ppk-Gal4* at WP and 16 hr APF. Red arrowheads point to the ddaC somas. (**A–C**) ddaC neurons co-expressing *patronin* RNAi #1 and control RNAi exhibited strong pruning defects at 16 hr APF (**A**). The pruning defects in *patronin* RNAi #1 ddaC neurons were significantly suppressed by knockdown of *klp10A* via either RNAi #1 (**B**) or RNAi #2 (**C**). (**F–H**) ddaC neurons co-expressing Patronin and the control
*Figure 7 continued on next page*

*Figure 7 continued*

RNAi construct (F) exhibited strong pruning defects at 16 hr APF. These pruning defects were significantly suppressed by knockdown of *klp10A* via either RNAi #1 (G) or RNAi #2 (H). (D, I) Quantification of total length of unpruned ddaC dendrites at 16 hr APF. (E, J) Quantification of the severing defects at 16 hr APF. Scale bar in (A) represents 50 μm. (K–L) Representative kymographs of EB1-GFP comets in control or mutant ddaC dendrites. In ddaC neurons co-overexpressing Patronin and the control RNAi construct, EB1-GFP comets moved bidirectionally in the dendrites (K). However, *klp10A* knockdown via RNAi #2 (L) significantly restored the retrograde movement of EB1-GFP comets in Patronin-overexpressing ddaC dendrites. Horizontal arrow indicates the direction towards the soma. Scale bar in (K) represents 10 μm, and each movie was taken for 3 min. (M) Quantification of percentage of dendritic EB1-GFP comets moving away from soma (anterogradely). (N) Quantification of average numbers of EB1-GFP comets along 30 μm of dendrites within 3 min. Error bars represent SEM. The number of samples (n) in each group is shown on the bars. N.S., not significant; **$p<0.01$; ***$p<0.001$ as assessed by one-way ANOVA test or two-tailed Student's T test.

DOI: https://doi.org/10.7554/eLife.39964.025

The following source data and figure supplements are available for figure 7:

**Source data 1.** Extended statistical data as Microsoft Excel spreadsheet.
DOI: https://doi.org/10.7554/eLife.39964.028

**Figure supplement 1.** *patronin* appears not to genetically interact with *kat-60*, *kat-60L1* or *tau* in dendrite pruning.
DOI: https://doi.org/10.7554/eLife.39964.026

**Figure supplement 1—source data 1.** Extended statistical data as Microsoft Excel spreadsheet.
DOI: https://doi.org/10.7554/eLife.39964.027

overexpression also significantly increased the percentage of anterograde EB1-GFP movement in the dendrites (11%, n = 16; *Figure 8G–H*), suggesting a dendritic MT orientation defect. Klp10A overexpression did not caused a signification reduction in the number of EB1-GFP comets in the dendrites (*Figure 8I*), resembling the *patronin* RNAi knockdown. In addition, neither track length of EB1-GFP comets nor their speed was affected upon Klp10A overexpression (*Figure 8J–K*). These data imply that *patronin* RNAi or Klp10A overexpression might result in excessive MT depolymerization primarily at the MT minus ends, rather than affect the number of EB1-GFP comets that mark MT growth at the MT plus ends. Thus, these data strongly demonstrate that Patronin promotes dendrite pruning via antagonising Klp10A function.

In addition, we examined the functional significance of the CKK domain of Patronin in regulating dendrite pruning and dendritic MT orientation. To this end, we first confirmed that the expression of the CKK domain alone significantly rescued the dendrite pruning defects in the *patronin* RNAi background at 16 hr APF (n = 24; *Figure 8—figure supplement 2A*), similar to those in *patronin*$^{c9-c5}$ ddaC neurons (*Figure 3D, E and F*). Importantly, the expression of the CKK domain, rather than the CH domain, almost fully rescued Nod-β-gal distribution (n = 15; *Figure 8—figure supplement 2B*) as well as retrograde EB1-GFP comets in the dendrites of *patronin* RNAi ddaC neurons (n = 15; *Figure 8—figure supplement 2C*). Moreover, the expression of the CKK domain significantly suppressed the dendrite pruning defects in Klp10A-overexpressing ddaC neurons (n = 30; *Figure 8—figure supplement 2D*), suggesting that the CKK domain is able to antagonize Klp10A's function during dendrite pruning. Thus, multiple lines of evidence demonstrate that the CKK domain is important for Patronin to govern minus-end-out MT orientation in dendrites as well as dendrite pruning.

In summary, Patronin orients uniform minus-end-out MT arrays in dendrites to facilitate dendrite pruning in ddaC sensory neurons during early metamorphosis.

## Discussion

During early metamorphosis, C4da or ddaC neurons undergo dendrite pruning to selectively eliminate their dendrites but keep their axons intact (*Williams and Truman, 2005*; *Kuo et al., 2005*). Dendritic MT breakdown precedes membrane scission, which leads to physical severing of proximal dendrites from the soma (*Williams and Truman, 2005*). In ddaC neurons, dendrites are enriched with MT minus ends and organized with predominant minus-end-out MT arrays, whereas the axons differ in MT orientation and acquire a plus-end-out pattern (*Rolls et al., 2007*; *Zheng et al., 2008*; *Satoh et al., 2008*). However, the involvement of MT minus-end-binding proteins in governing dendritic MT orientation and thereby dendrite pruning is completely unknown. In this study, we identified a MT minus-end-binding protein Patronin for its key role in dendrite-specific pruning of ddaC

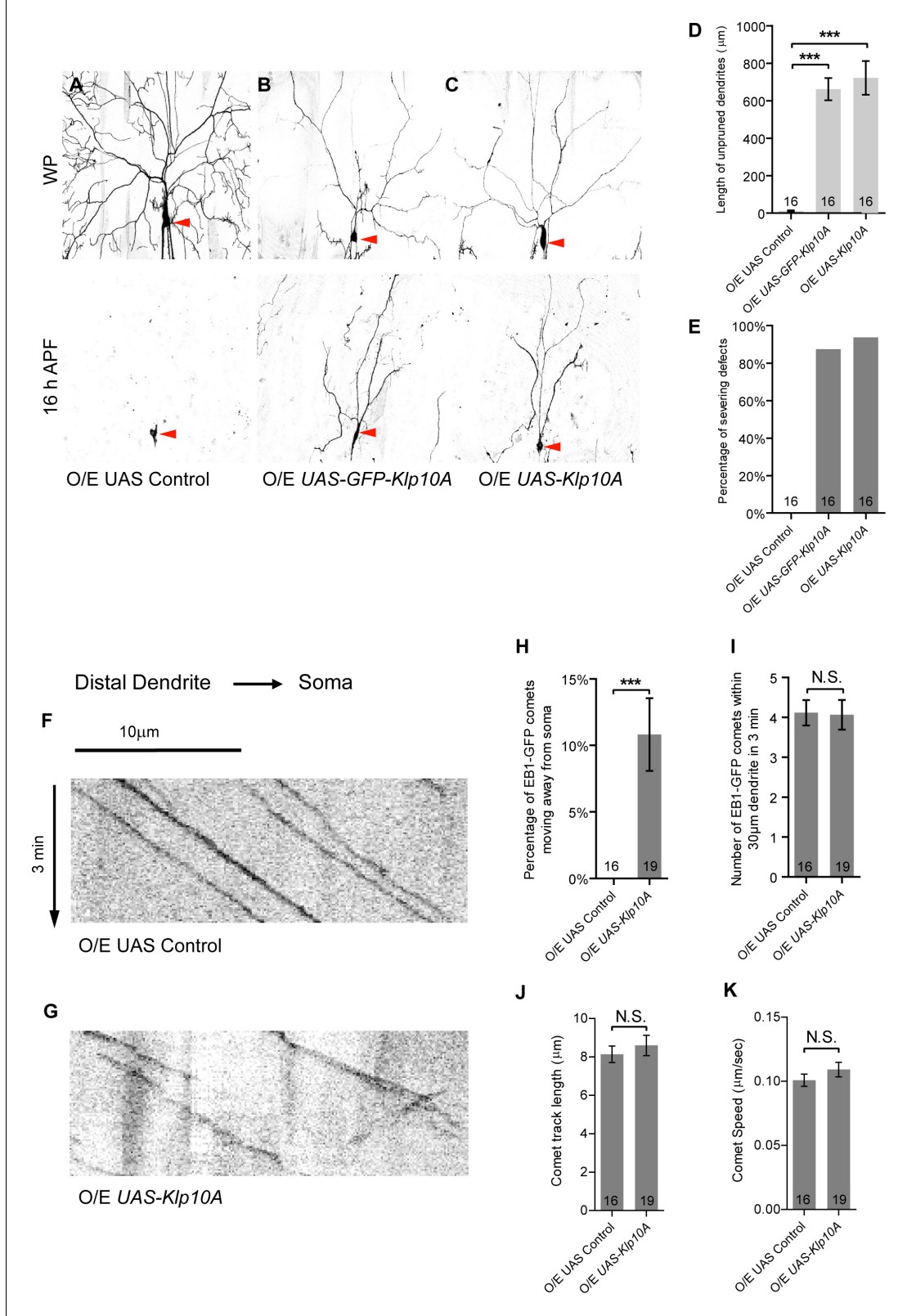

**Figure 8.** Klp10A overexpression phenocopies *patronin* knockdown in dendrite pruning and dendritic MT orientation. (**A–C**) Live confocal images of ddaC neurons expressing mCD8-GFP driven by *ppk-Gal4* at WP and 16 hr APF. Red arrowheads point to the ddaC somas. ddaC neurons overexpressing the *UAS* control construct pruned all the dendrites (**A**), whereas ddaC neurons overexpressing GFP-Klp10A (**B**) or Klp10A (**C**) via two copies of *ppk-Gal4* exhibited simple arbors at WP stage and dendrite pruning defects at 16 hr APF. (**D**) Quantification of total length of unpruned ddaC

*Figure 8 continued on next page*

*Figure 8 continued*

dendrites at 16 hr APF. (E) Quantification of the severing defects at 16 hr APF. Scale bar in (A) represents 50 μm. (F–G) Representative kymographs of EB1-GFP comets in *UAS* control or Klp10A-overexpressing ddaC dendrites. In the dendrites of the control ddaC neurons (F), EB1-GFP comets moved towards the somas. However, in ddaC neurons overexpressing Klp10A (G), EB1-GFP comets moved bidirectionally in the dendrites. Horizontal arrow indicates the direction towards the soma. Scale bar in (F) represents 10 μm, and each movie was taken for 3 min. (H) Quantification of the percentage of EB1-GFP comets moving away from soma (anterogradely) in ddaC dendrites. (I) Quantification of average numbers of EB1-GFP comets along 30 μm of dendrites within 3 min. (J) Quantification of average track length of EB1-GFP comets along 30 μm of dendrites within 3 min. (K) Quantification of average speed of EB1-GFP comets. Error bars represent SEM. The number of samples (n) in each group is shown on the bars. N.S., not significant; ***p<0.001 as assessed by one-way ANOVA test or two-tailed Student's T test.

DOI: https://doi.org/10.7554/eLife.39964.029

The following source data and figure supplements are available for figure 8:

**Source data 1.** Extended statistical data as Microsoft Excel spreadsheet.
DOI: https://doi.org/10.7554/eLife.39964.036
**Figure supplement 1.** Overexpression of Klp10A causes dendritic MT orientation defects in ddaC neurons.
DOI: https://doi.org/10.7554/eLife.39964.030
**Figure supplement 1—source data 1.** Extended statistical data as Microsoft Excel spreadsheet.
DOI: https://doi.org/10.7554/eLife.39964.031
**Figure supplement 2.** The CKK domain of Patronin is important for dendrite pruning and dendritic MT orientation.
DOI: https://doi.org/10.7554/eLife.39964.032
**Figure supplement 2—source data 1.** Extended statistical data as Microsoft Excel spreadsheet.
DOI: https://doi.org/10.7554/eLife.39964.033
**Figure supplement 3.** *CNN* and *APC1/2* appear to be required for dendrite pruning of ddaC neurons.
DOI: https://doi.org/10.7554/eLife.39964.034
**Figure supplement 3—source data 1.** Extended statistical data as Microsoft Excel spreadsheet.
DOI: https://doi.org/10.7554/eLife.39964.035

neurons. Patronin/CAMSAP/PTRN-1 were previously reported to track and stabilize MT minus ends, and regulate axon specification and dendrite morphology in mammals and worms (*Richardson et al., 2014*; *Yau et al., 2014*; *Marcette et al., 2014*), but their roles in neuronal MT orientation remained unknown. In this study, we provide the evidence indicating that Patronin plays an important role in governing minus-end-out orientation of dendritic MTs in *Drosophila* ddaC sensory neurons. Mechanistically, we show that Patronin regulates dendritic MT orientation to facilitate dendrite pruning by suppressing the Klp10A activity. Thus, we demonstrate, for the first time, that a MT minus-end-binding protein facilitates neuronal pruning by orienting proper MT arrays in neurites.

## Patronin is a novel player of dendrite pruning in ddaC neurons

Patronin was first identified as a regulator of mitotic spindle (*Goshima et al., 2007*). Patronin recognizes and stabilizes free MT minus ends against kinesin 13-mediated MT depolymerization in cultured S2 cells (*Goodwin and Vale, 2010*). Recent studies have started to unravel the important roles of Patronin in neuronal development and differentiation. Mammalian CAMSAP2 stabilizes neuronal MTs in axons and dendrites by its association with noncentrosomal MT minus ends; it is required for axon specification and dendrite arborization (*Yau et al., 2014*). In *C. elegans*, PTRN-1 is required for several developmental processes, such as synapse stabilization and neurite formation (*Marcette et al., 2014*; *Richardson et al., 2014*). In this study, we provide multiple lines of in vivo evidence to demonstrate that Patronin is a novel regulator of dendrite pruning in ddaC neurons, and Patronin regulates dendrite pruning by suppressing the activity of Klp10A. First, genetic analyses with multiple RNAi lines, two *patronin* mutants as well as the rescue experiment unambiguously reveal that Patronin is required for dendrite pruning in a cell-autonomous manner. Second, we conducted gain-of-function studies using two independent transgenes and further demonstrated that Patronin acts in a dose-sensitive manner to govern dendrite pruning. Third, we also show Patronin's role in initial dendrite arborization, which, however, is separable from its function in dendrite pruning. Fourth, attenuation of Klp10A function significantly rescued the *patronin* mutant phenotype in terms of dendrite pruning. Finally, overexpression of Klp10A inhibited dendrite pruning and resembled the *patronin* loss-of-function mutants. Thus, Patronin promotes dendrite pruning primarily via antagonizing the Klp10A activity.

Patronin/CAMSAPs/PTRN-1 contains a N-terminal CH domain, three CC domains, and a MT-binding CKK domain at the C-terminal region (*Baines et al., 2009*). Mammalian CAMSAPs recognize MT minus-ends via their CKK domains (*Jiang et al., 2014*). *Drosophila* Patronin was reported to target MT minus ends via its CC domains, whereas its CKK domain localizes along MTs and is essential for supressing MT minus-end dynamics in *Drosophila* S2 cells (*Goodwin and Vale, 2010*; *Hendershott and Vale, 2014*). In *C. elegans* neurons, the CKK domain of PTRN-1 is necessary and sufficient for its function in MT dynamics and axon regeneration (*Chuang et al., 2014*). Our functional domain analyses reveal that the MT-binding CKK domain is important for Patronin's function in dendrite pruning of ddaC neurons, whereas CH and CC domains appear to be less important. Thus, our study further highlights a functional importance of the CKK domain in neuronal pruning. It is conceivable that Patronin acts via this domain to stabilize dendritic MT minus ends and thereby orient normal dendritic MTs to facilitate dendrite-specific pruning of ddaC neurons. Consistent with this idea, a recent structural analysis revealed that the CKK domain preferentially binds to a curved region of MT minus ends on the outer surface to prevent the association of kinesin-13 depolymerase with the same site via steric inhibition, thereby protecting MT minus ends from kinesin-13-mediated MT depolymerization (*Atherton et al., 2017*).

## Patronin regulates the minus-end-out orientation of dendritic MTs in ddaC dendrites

Mammalian CAMSAP3/Nezha interacts with the spectraplakin ACF7, a cytoskeletal crosslinking protein, anchors non-centrosomal MT minus ends to actin filaments and thereby polarizes MT networks in cultured epithelial cells (*Ning et al., 2016*; *Noordstra et al., 2016*). However, it remains unknown about a role of Patronin/CAMSAPs/PTRN-1 in regulating MT polarity in differentiated neurons. Here, we demonstrate that Patronin governs uniform MT orientation in the major dendrites of ddaC sensory neurons. We first show that the MT minus-end marker Nod-β-gal was greatly reduced in the dendrites of *patronin* mutant neurons, concomitantly with its robust accumulation in the soma. This finding suggests a strong reduction in non-centrosomal minus ends in the dendrites. Moreover, the axon-specific marker Kin-β-gal, which marks MT plus ends, mis-localized to the dendrites in *patronin* mutant neurons, indicative of increased plus-end-out MT arrays in the dendrites. Consistent with these observations, we further show a significant increase in anterior EB1-GFP comets in the major dendrites of *patronin* neurons, suggesting impaired MT orientation. Moreover, the number of EB1-GFP comets, which reflects overall MT density and MT nucleation levels in the dendrites, was not significantly different between wild-type and *patronin* neurons. Finally, we demonstrate that Patronin controls MT minus-end-out orientation by suppressing the kinesin-13/Klp10A activity.

How might Patronin regulate MT minus-end-out orientation in ddaC dendrites? Similar to its mammalian counterpart in cultured hippocampal neurons, Patronin likely associates in vivo with non-centrosomal MT minus-ends and stabilizes them against kinesin-13-mediated MT depolymerization in the ddaC dendrites. Via MT guidance or sliding by plus-end motors kinesins, growing MTs might be oriented in a minus-end-out manner in the dendrites (*Yan et al., 2013*; *Mattie et al., 2010*). In the absence of Patronin, dendritic MTs might be depolymerized into short filaments from their minus ends mediated by excessive depolymerization activity of Klp10A and/or other MT severing factors. Microtubule depolymerising or severing factors have been observed to depolymerize MTs into short filaments (*McNally and Vale, 1993*). Short MT filaments, which was proposed to be re-oriented in either plus-end-out or minus-end-out direction with equal probability (*del Castillo et al., 2015*), can serve as seeds for MT growth, resulting in a mixed MT polarity in the dendrites of *patronin* mutant neurons. Consistent with this speculation, we show that the MT minus-end marker Nod-β-gal robustly accumulated in the soma of *patronin* mutant neurons. Moreover, depletion of Klp10A in *patronin* mutant neurons fully restored dendritic distribution of the minus-end-marker Nod-β-gal as well as the minus-end-out MT orientation in the dendrites, supporting the idea that kinesin-13-dependent MT depolymerization at the minus ends is attributed to the impaired MT orientation in the dendrites. We further show that the number of EB1-GFP comets was not significantly different between wild-type and *patronin* neurons, suggesting that MT nucleation levels or plus-end growth activity appears to be unaffected by the absence of Patronin. *patronin* depletion might result in excessive MT depolymerization primarily at the MT minus ends. Consistent with this notion, it was reported that in Patronin-deficient *Drosophila* S2 cells, minus end depolymerization often halted when it reached the EB1-enriched MT plus end tips, indicating that +TIP proteins might resist

continued minus-end depolymerization (*Goodwin and Vale, 2010*). Second, given that a small proportion of CAMSAP3 is able to bind to MT plus ends in cultured epithelial cells (*Ning et al., 2016*), one might envisage that Patronin might bind to dendritic MT plus ends in ddaC neurons and regulate MT orientation together with the MT plus-end-binding proteins such as kinesin-1/2, EB-1 and APC2. These proteins were previously reported to regulate dendritic MT orientation (*Mattie et al., 2010*; *Herzmann et al., 2018*; *Yan et al., 2013*); kinesin-1/2 and EB-1 have recently shown to be involved in dendrite pruning although the mechanism remains unknown (*Herzmann et al., 2018*) (Wang and Yu, unpublished data). The potential relevance between Patronin and plus-end regulators awaits further investigation in future. Finally, Patronin is recruited to the actin-rich anterior cortex by Shot, a cytoskeletal crosslinking protein, and controls the anterior-posterior MT polarity in *Drosophila* oocytes (*Nashchekin et al., 2016*). Patronin might also anchor MT minus ends to actin through Shot in ddaC sensory neurons. Given that the distal dendrites of ddaC neurons are actin-rich (*Medina et al., 2006*; *Nagel et al., 2012*), Patronin might tether free minus ends of MT filaments to the distal branches to elongate dendritic MTs with minus-end-out orientation. This possibility could be interrogated by removal of Shot or actin regulators in ddaC neurons in future studies.

## A high correlation between MT orientation and dendrite-specific pruning in ddaC sensory neurons

Neuron highly relies on its MT cytoskeleton to support neuronal architecture as well as facilitate intracellular transport of proteins and organelles. In mammalian neurons, axonal MTs are oriented plus-end-out, whereas dendritic MTs have mixed orientations with both plus-end-out and minus-end-out patterns (*Baas and Lin, 2011*; *Yau et al., 2016*). MTs are arranged in a predominant minus-end-out orientation in major dendrites of *Drosophila* and *C. elegans* neurons (*Stone et al., 2008*; *Goodwin et al., 2012*). In *Drosophila*, developing da neurons, including ddaC neurons, initially exhibit a mixed orientation of MTs in dendrites and gradually mature to have a uniform minus-end-out pattern over several days (*Hill et al., 2012*). Mature ddaC neurons maintain the minus-end-out orientation of MTs in their dendrites before the onset of pruning (Wang and Yu, unpublished data). Emerging evidence suggests that uniform dendritic MT orientation might be a prerequisite for dendrite pruning. First, in axon-injured da neurons, dendritic MT orientation becomes mixed instead of minus-end-out (*Stone et al., 2010*; *Song et al., 2012*); concurrently, dendrite pruning is also inhibited in ddaC neurons following axon transection (*Chen et al., 2012*). Second, *kinesin-1/2* mutant ddaC neurons show both mixed dendritic MT orientation and dendrite pruning defects (*Stone et al., 2010*; *Herzmann et al., 2018*). In this study, we further identified the MT minus-end-binding protein Patronin that is required for dendrite pruning of ddaC neurons, and loss of *patronin* function led to decreased Nod-β-gal levels and increased anterograde EB1-GFP comets in the dendrites. More importantly, co-depletion of the kinesin-13 MT depolymerase Klp10A restored normal Nod-β-gal levels and uniform minus-end-out MT orientation in *patronin* mutant dendrites and thereby rescued the dendrite pruning defects. In addition, CNN and APC1/2, two known regulators of dendritic MT polarity (*Yalgin et al., 2015*; *Mattie et al., 2010*), also appear to be required for dendrite pruning. We observed dendrite severing defects in *cnn*[hk1] mutant ddaC neurons (52%, n = 23; *Figure 8—figure supplement 3A*) as well as *APC1*[Q8], *APC2*[N175K] double mutant ddaC clones (60%, n = 10; *Figure 8—figure supplement 3B*). As controls, MT-associated proteins, Futsch and Tau, which regulate MT stability (*Hummel et al., 2000*; *Herzmann et al., 2017*), appear to be dispensable for dendrite pruning, as no dendrite pruning defect was observed in *fustch*[N94] or *tau* RNAi ddaC neurons (*Figure 8—figure supplement 3C–D*). Thus, multiple lines of evidence demonstrate that dendritic MT orientation is important for dendrite pruning. Our study favors a model that Patronin stabilizes MT minus ends and maintains MT minus-end-out orientation in dendrites to facilitate the MT breakdown and thereby dendrite pruning. It is possible that uniform minus-end-out MTs may provide the tracks for localizing MT severing factors at the proximal dendrites to disassemble local MT filaments.

## Materials and methods

**Key resources table**

*Continued on next page*

*Continued*

| Reagent type | Designation | Source or reference | Identifiers | Additional information |
|---|---|---|---|---|
| Reagent type | Designation | Source or reference | Identifiers | Additional information |
| Genetic reagent (D. melangoaster) | UAS-Mical$^{N-ter}$ | other | | (*Terman et al., 2002*) |
| Genetic reagent (D. melangoaster) | SOP-flp (#42) | other | | (*Matsubara et al., 2011*) |
| Genetic reagent (D. melangoaster) | ppk-Gal4 on II and III chromosome | other | | (*Grueber et al., 2003*) |
| Genetic reagent (D. melangoaster) | UAS-Kin-$\beta$-gal | other | | (*Clark et al., 1997*) |
| Genetic reagent (D. melangoaster) | UAS-EB1-GFP | other | | (*Stone et al., 2008*) |
| Genetic reagent (D. melangoaster) | UASp-mCherry-Patronin | other | | (*Nashchekin et al., 2016*) |
| Genetic reagent (D. melangoaster) | patronin$^{c9-c5}$ | other | | (*Nashchekin et al., 2016*) |
| Genetic reagent (D. melangoaster) | UAS-GFP-Patronin | other | | (*Derivery et al., 2015*) |
| Genetic reagent (D. melangoaster) | UASp-Arf79F-EGFP | other | | (*Shao et al., 2010*) |
| Genetic reagent (D. melangoaster) | UAS-Klp10A | this paper | | |
| Genetic reagent (D. melangoaster) | UAS-GFP-Klp10A | this paper | | |
| Genetic reagent (D. melangoaster) | UAS-Patronin | this paper | | |
| Genetic reagent (D. melangoaster) | UAS-Venus-Patronin | this paper | | |
| Genetic reagent (D. melangoaster) | UASp-Patronin$^{\Delta CH}$ | this paper | | |
| Genetic reagent (D. melangoaster) | UASp-Patronin$^{\Delta CKK}$ | this paper | | |
| Genetic reagent (D. melangoaster) | UASp-Patronin$^{CKK}$ | this paper | | |
| Genetic reagent (D. melangoaster) | UAS-Venus-Patronin$^{\Delta CH}$ | this paper | | |
| Genetic reagent (D. melangoaster) | UAS-Venus-Patronin$^{\Delta CKK}$ | this paper | | |
| Genetic reagent (D. melangoaster) | UAS-Venus-Patronin$^{CC1-3}$ | this paper | | |
| Genetic reagent (D. melangoaster) | UAS-Venus-Patronin$^{CKK}$ | this paper | | |
| Genetic reagent (D. melangoaster) | UAS-Venus-Patronin$^{CH}$ | this paper | | |
| Genetic reagent (D. melangoaster) | Gal4$^{109(2)80}$ | Bloomington Stock Center | BDSC: 8769 | |
| Genetic reagent (D. melangoaster) | ppk-CD4-tdGFP | Bloomington Stock Center | BDSC: 35843 | |
| Genetic reagent (D. melangoaster) | GSG2295-Gal4 | Bloomington Stock Center | BDSC: 40266 | |
| Genetic reagent (D. melangoaster) | Gal4$^{4-77}$ | Bloomington Stock Center | BDSC: 8737 | |

*Continued on next page*

Continued

| Reagent type | Designation | Source or reference | Identifiers | Additional information |
|---|---|---|---|---|
| Genetic reagent (*D. melangoaster*) | *UAS-Nod-β-gal* | Bloomington Stock Center | BDSC: 9912 | |
| Genetic reagent (*D. melangoaster*) | *patronin* RNAi #2 | Bloomington Stock Center | BDSC: 36659 | |
| Genetic reagent (*D. melangoaster*) | *klp10A* RNAi #2 | Bloomington Stock Center | BDSC: 33963 | |
| Genetic reagent (*D. melangoaster*) | *Gal4*[2-21] | Bloomington Stock Center | FBal0328157 | |
| Genetic reagent (*D. melangoaster*) | *patronin* RNAi # 1 | Vienna *Drosophila* RNAi Centre | VDRC: v108927 | |
| Genetic reagent (*D. melangoaster*) | *klp10A* RNAi # 1 | Vienna *Drosophila* RNAi Centre | VDRC: v41534 | |
| Genetic reagent (*D. melangoaster*) | control RNAi | Vienna *Drosophila* RNAi Centre | VDRC: v36355 | |
| Genetic reagent (*D. melangoaster*) | control RNAi | Vienna *Drosophila* RNAi Centre | VDRC: v37288 | |
| Genetic reagent (*D. melangoaster*) | *patronin*[k07433] | Drosophila Genetic Resource Center | #111217 | |
| Genetic reagent (*D. melangoaster*) | *patronin* RNAi #3 | National Institute of Genetics, Japan | #18462 Ra-1 | |
| Genetic reagent (*D. melangoaster*) | *kat-60* RNAi # 1 | Vienna *Drosophila* RNAi Centre | VDRC: v38368 | |
| Genetic reagent (*D. melangoaster*) | *kat-60* RNAi # 2 | Vienna *Drosophila* RNAi Centre | VDRC: v106487 | |
| Genetic reagent (*D. melangoaster*) | *kat-60L1* RNAi # 1 | Vienna *Drosophila* RNAi Centre | VDRC: v31599 | |
| Genetic reagent (*D. melangoaster*) | *kat60-L1* RNAi # 2 | Vienna *Drosophila* RNAi Centre | VDRC: v108168 | |
| Genetic reagent (*D. melangoaster*) | *tau* RNAi # 1 | Bloomington Stock Center | BDSC: 28891 | |
| Genetic reagent (*D. melangoaster*) | *tau* RNAi # 2 | Bloomington Stock Center | BDSC: 40875 | |
| Genetic reagent (*D. melangoaster*) | *cnn*[hk21] | Bloomington Stock Center | BDSC: 5039 | |
| Genetic reagent (*D. melangoaster*) | *FRT 82B, APC2*[N175K], *APC1*[Q8] | Bloomington Stock Center | BDSC: 7211 | |
| Genetic reagent (*D. melangoaster*) | *futsch*[N94] | Bloomington Stock Center | BDSC: 8805 | |
| Antibody | anti-β-galactosidase | Promega | Cat#: Z3783 | 1:1000 |
| Antibody | anti-Patronin | M Gonzalez-Gaitan | | 1:500 |
| Antibody | Cy3-conjugated goat anti-Rabbit antibody | Jackson | Cat#111-165-003 | 1:500 |
| Antibody | 647-conjugated goat anti-Rabbit antibody | Jackson | Cat#111-605-144 | 1:500 |
| Antibody | anti-Futsch | DSHB | 22c10 | 1:50 |

## Fly strains

*UAS-Mical*[N-ter] (**Terman et al., 2002**), *SOP-flp* (#42) (**Matsubara et al., 2011**), *ppk-Gal4* on II and III chromosome (**Grueber et al., 2003**), *UAS-Kin-β-gal* (**Clark et al., 1997**), *UAS-EB1-GFP* (**Stone et al., 2008**), *UASp-mCherry-Patronin*, *patronin*[c9-c5] (**Nashchekin et al., 2016**), *UAS-GFP-Patronin* (**Derivery et al., 2015**), *UASp-Arf79F-EGFP* (**Shao et al., 2010**), *UAS-Klp10A*, *UAS-GFP-Klp10A*, *UAS-Patronin, UAS-Venus-Patronin, UASp-Patronin*[ΔCH], *UASp-Patronin*[ΔCKK], *UASp-Patronin*[CKK], *UAS-*

*Venus-Patronin$^{\Delta CH}$*, *UAS-Venus-Patronin$^{\Delta CKK}$*, *UAS-Venus-Patronin$^{CC1-3}$*, *UAS-Venus-Patronin$^{CKK}$*, *UAS-Venus-Patronin$^{CH}$* (this study).

The following stocks were obtained from Bloomington Stock Center (BSC): *Gal4$^{109(2)80}$, ppk-CD4-tdGFP* (BL#35843), *GSG2295-Gal4* (BL#40266), *Gal4$^{4-77}$* (BL#8737), *UAS-Nod-β-gal* (BL#9912), *patronin* RNAi #2 (BL#36659), *klp10A* RNAi #2 (BL#33963), *Gal4$^{2-21}$, tau* RNAi #1 (BL#28891) and #2 (BL#40875), *cnn$^{hk21}$* (BL#5039), *APC2$^{N175K}$, APC1$^{Q8}$* (BL#7211), *futsch$^{N94}$* (BL#8805).

The following stocks were obtained from Vienna *Drosophila* RNAi Centre (VDRC): *patronin* RNAi # 1 (v108927), *klp10A* RNAi # 1 (v41534), control RNAi (v36355, v37288), *kat-60* RNAi #1 (v38368) and #2 (v106487), *kat-60L1* RNAi #1 (v31599) and #2 (v108168).

The following stock was obtained from Drosophila Genetic Resource Center (DGRC), Kyoto: *patronin$^{k07433}$* (#111217).

The following stock was obtained from National Institute of Genetics, Japan: *patronin* RNAi #3 (#18462 Ra-1).

## Generation of *patronin* and *klp10A* Transgenes

*patronin* and *klp10A* full-length cDNAs were PCR amplified from pMT-mCherry-Patronin (Addgene) and EST LD29208 (DGRC, Bloomington) into *pDonor* vector (Life Tech). The GATEWAY *pTW, pTVW* or *pTGW* vectors (DGRC) containing the respective fragments of the cDNAs were constructed by LR reaction (Life Tech).

The variants of *patronin* were generated by either PCR or site mutagenesis (Agilent Tech) using *pDonor patronin* as a template. The respective cDNA fragments were subcloned into *pTVW* or *pPW* vector (DGRC). The transgenic lines were established by the Bestgene Inc.

## Immunohistochemistry and antibodies

The following primary antibodies were used for immunohistochemistry at the indicated dilution: mouse anti-β-galactosidase (Promega Cat#: Z3783, 1:1000), Rabbit anti-Patronin (a gift from M. Gonzalez-Gaitan, 1:500), mouse anti-Futsch (22C10, DSHB, 1:50). Cy3 or Alexa Fluor 647-conjugated secondary antibodies (Jackson Laboratories, Cat#: 111-165-003, 111-605-144) were used at 1:500 dilution. For immunostaining, pupae or larvae were dissected in PBS and fixed with 4% formaldehyde for 15 min. Mounting was performed in VectaShield mounting medium, and the samples were directly visualized by confocal microscopy.

## Live imaging analysis

To image *Drosophila* da neurons at the wandering 3$^{rd}$ instar (wL3) or WP stage, larvae or pupae were first washed in PBS buffer briefly and followed by immersion with 90% glycerol. For imaging da neurons at 16 hr APF or 20 hr APF, pupal cases were carefully removed before mounted with 90% glycerol. Dendrite images were acquired on Leica TSC SP2.

## MARCM analysis of da sensory neurons

MARCM analysis, dendrite imaging, and quantification were carried out as previously described. ddaC clones were selected and imaged at the WP stage according to their location and morphology. The ddaC neurons were examined for dendrite pruning defects at 16 hr APF.

## RU486/mifepristone treatment for the Gene-Switch system

RU486/mifepristone treatment for the Gene-Switch system was carried out as previously described. Embryos were collected at 6 hr intervals and reared on standard food to the early 3rd instar larva stage. The larvae were transferred to the standard culture medium which contains 240 µg/ml mifepristone (Sigma Aldrich M8046). White prepupae were picked up, subject to phenotypic analysis at WP or 16 hr APF.

## Quantification of ddaC dendrites

Live confocal images of ddaC neurons expressing UAS-mCD8-GFP driven by *ppk-Gal4* or *Gal4$^{4-77}$*, or *ppk-CD4-tdGFP* were shown at WP, 16 hr APF. For wild-type or mutant ddaC neurons, the percentages of severing defect were quantified in a 275 µm x 275 µm region of the dorsal dendritic field, originating from the abdominal segments 2–5. The severing defect was defined by the

presence of dendrites that remain attached to the soma at 16 hr APF (*Kirilly et al., 2009*; *Williams and Truman, 2005*; *Kuo et al., 2005*). Total length of unpruned dendrites was measured in a 275 μm x 275 μm region of the dorsal dendritic field using ImageJ. The number of samples (n) in each group is shown on the bars. Statistical significance was determined using either two-tailed Student's t-test (two samples) or one-way ANOVA and Bonferroni test (multiple samples) (*$p<0.05$, **$p<0.01$, ***$p<0.001$, N.S., not significant). Error represent S.E.M. Dorsal is up in all images.

### EB1-GFP comet imaging

Wild-type and mutant embryos were collected at 3 hr intervals. Embryos were reared on standard cornmeal food without yeast supplement until 96 hr AEL, followed by EB1-GFP imaging. EB1-GFP comet imaging was performed with Olympus FV3000 using 60X Oil lens and Zoom factor 3. ddaC neurons of 96 hr AEL larva were imaged for 3 min. 82 frames were acquired for each neuron at 2.25 s interval and analyzed using and ImageJ software.

## Acknowledgements

We thank M Gonzalez-Gaitan, TJ Harris, YN Jan, D St Johnston, AL Kolodkin, MM Rolls, T Uemura, the Bloomington Stock Center (BSC), VDRC (Austria), *Drosophila* Genetic Resource Center (Japan), National Institute of Genetics (Japan) for generously providing antibodies and fly stocks. We are grateful to MM Rolls for providing the technical advice on EB1-GFP imaging. We thank members of the FY laboratory for helpful assistance. We also thank the TLL microscopy facility for technical assistance. YW was recipients of the NGS postgraduate scholarship, Singapore. This work was supported by Temasek Life Sciences Laboratory (TLL), Singapore (to FY).

## Additional information

### Funding

| Funder | Grant reference number | Author |
|---|---|---|
| Temasek Life Sciences Laboratory, Singapore | TLL-2040 | Fengwei Yu |
| National Research Foundation, Singapore | SBP-P3, SBP-P8 | Fengwei Yu |

The funders had no role in study design, data collection and interpretation, or the decision to submit the work for publication.

### Author contributions

Yan Wang, Resources, Data curation, Software, Formal analysis, Investigation, Visualization, Methodology, Writing—original draft; Menglong Rui, Resources, Data curation, Formal analysis, Investigation, Visualization, Methodology; Quan Tang, Resources, Data curation, Software, Formal analysis, Visualization, Methodology; Shufeng Bu, Formal analysis, Investigation, Visualization; Fengwei Yu, Conceptualization, Resources, Data curation, Formal analysis, Supervision, Funding acquisition, Validation, Investigation, Visualization, Writing—original draft, Project administration, Writing—review and editing

### Author ORCIDs

Fengwei Yu http://orcid.org/0000-0003-0268-199X

### Decision letter and Author response

Decision letter https://doi.org/10.7554/eLife.39964.040
Author response https://doi.org/10.7554/eLife.39964.041

## Additional files

### Supplementary files
• Supplementary file 1. Supplementary list of fly strains.
DOI: https://doi.org/10.7554/eLife.39964.037
• Transparent reporting form
DOI: https://doi.org/10.7554/eLife.39964.038

### Data availability
All data generated or analysed during this study are included in the manuscript and supporting files.

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
