## [Decision Letter]

[**Editorial note:** This article has been through an editorial process in which the authors decide how to respond to the issues raised during peer review. The Reviewing Editor's assessment is that minor issues remain unresolved.]

Evaluation of the final submission:

Minor issues remain unresolved. In particular, the explanation of why both loss and gain of Patronin lead to pruning defects is not definitive.

Decision letter after peer review:

Thank you for submitting your article "Patronin governs minus-end-out orientation of dendritic microtubules to promote dendrite pruning in *Drosophila*" for consideration by *eLife*. Your article has been reviewed by three peer reviewers, and the evaluation has been overseen by a Reviewing Editor and Anna Akhmanova as the Senior Editor. The following individuals involved in review of your submission have agreed to reveal their identity: Shaul Yogev (Reviewer #1). The other two reviewers remain anonymous.

The Reviewing Editor has highlighted the concerns that require revision and/or responses, and we have included the separate reviews below for your consideration. If you have any questions, please do not hesitate to contact us.

Summary:

All three reviewers found your manuscript addressing an important question in neuronal cell biology. They also agree that this work contains many interesting data that are certainly of interest to the general readership of *eLife*. However, a number of important concerns were raised. I would like to ask you to address these concerns with additional experiments and discussion.

Major concerns:

1) Reviewers found that the causality between MT polarity defects and pruning is lacking. We suggest you to examine other mutants with MT polarity defects like cnn, APC to see if they have pruning phenotypes.

2) The authors hypothesized that the defects observed in Patronin knockdown are due to excessive depolymerization by *klp10a*. However, the number of EB1 comets is the same. It is just comet directionality that is perturbed. If the excessive depolymerization by *klp10a* is the cause of the phenotype, then there should be far fewer comets. It would make sense if *klp10a* were a cross-linking or sliding kinesin, but a depolymerising kinesin only having an effect on orientation and not number of EB1 comets seems bizarre. Why would misorientation be a result of excessive depolymerization activity? This is an important question that needs to be addressed because it gets at the fundamental question on the relationship between MT stability and polarity.

3) Genetic interaction between Patronin and Katanin *kat60L* should be addressed.

4) Please address whether Patronin overexpression affects dendrite pruning through the same mechanism as the loss-of-function mutant. This overexpression effect is not consistent with the model provided.

5) The CKK domain of Patronin is necessary and sufficient for pruning, which seems to be a surprising result considering that it is the minimal domain necessary to bind the MT. How can one explain this result?

We would also like to invite you to respond to the minor points that are listed in the individual reviews.

Separate reviews (please respond to each point):

*Reviewer #1:*

This is a very interesting paper that describes an unexpected role for Patronin in controlling dendrite MT polarity and pruning in *Drosophila*. The experiments are overall convincing and are properly controlled and the topic and scope are fit for *eLife*.

My main concern is that although this paper makes a very interesting connection between microtubule stability, polarity and pruning, it is not clear from the experiments how stabilization of MTs favors minus-end-out orientation, and whether it is polarity or stability that is important for pruning. This latter point could be addressed by examining some of the other mutants that affect MT polarity (cnn, APC, EB or others) or stability (Futch, Tau or others). by using genetic interactions the authors could place the Patronin-KLP10 module in the context of what we already know about MT regulation in this system.

My second reservation is that I am not convinced that Patronin overexpression affects dendrite pruning through the same mechanism as the loss-of-function as the paper seems to imply. This is because OE of the CKK domain has no effect and because loss of KLP10 (which should be equivalent to Patronin overexpression) has no phenotype. The authors should clearly discuss the possible distinct mechanisms of Patronin loss and gain of function.

Minor Comments:

– Please show more representative kymographs for the EB movies (i.e that show that most of the time the phenotype is not a complete reversal), and move the quantification from the supplementary figure to main figure.

– Please quantify MT growth speed and run length for EB movies. This could help to understand whether there are different properties to the minus-end-out versus plus-end-out nucleation.

– Is there any effect on axonal MT polarity (EB movies) in Patronin loss of function or KLP10 OE?

– Please add tubulin staining to support the conclusion that overall MT levels are not changed.

– What happens at later time-points? Do the mutant dendrites eventually "catch-up"?

– I was confused by the fact that the Patronin phenotype is 100% penetrant with the overexpressed motor fusions but only partial with the EB comets. Does this mean that these motor constructs "read" something besides MT polarity? Please comment on this.

*Reviewer #2:*

In their manuscript, Wang, Rui and Yu investigate the role of the microtubule (MT) minus-end binding protein, Patronin, in *Drosophila* dendritic arborization neurons. The open questions the authors sought to address were: 1) the role of Patronin in neuronal development and pruning, and 2) the contribution of the minus-end out MT orientation to dendrite pruning. Both knockdown and overexpression of Patronin inhibits the normal timecourse of dendritic pruning in the *Drosophila* pupal dendritic arborization neurons. They find that Patronin is necessary for the maintenance of MT polarity in dendrites. Instead of microtubules being predominantly minus-end out, in Patronin knockdown or mutant neurons, there is a mixture of minus- and plus-end out MTs. Inhibiting the MT depolymerizing kinesein, *klp10A* (kinesin-13), in this background restores the polarity of the MTs, indicating that in the absence of *patronin, klp10A* somehow contributes to this mixed MT polarity. Inhibiting *klp10a* also rescues the dendritic pruning defects observed in the absence of Patronin. The overall message of the paper is that Patronin antagonizes *klp10a* activity normally to maintain uniform MT minus-end out polarity during dendritic maintenance and pruning. In the absence of *patronin*, klp10A somehow functions in disrupting the uniform polarity of microtubules in the dendrites, which in turn disrupts dendritic pruning. The data are very nice and the results are clear; however, the model is incomplete. The effect of Patronin and *klp10a* on MTs are correlative with the pruning defects, not causative, and there can be a wide variety of other MAPs (stabilizers, severing proteins, etc.) whose functions are directly or indirectly disrupted upon Patronin or *klp10a* knockdown. These other possibilities are completely ignored, making the model disjointed and speculative.

The authors do not mention the direct interaction between Patronin and the MT severing enzyme, katanin (Jiang et al., 2014 and Jiang et al., 2018, the latter of which was not even cited). The authors do mention that katanin itself does not have a role in pruning, but katanin p60-like does based on prior work. It seems important to look at the genetic interaction between Patronin and Kat60L1 as well, before constructing an entire model around Patronin and *klp10a*.

The authors state many times that Patronin may stabilize uniform minus-end out MTs in the dendrites, but do not explain why and are unable to get at a mechanism in their paper. If the authors think that Patronin anchors MT minus ends in some way in the dendrites, then the CKK results do not make sense. The CKK domain of Patronin is necessary and sufficient for pruning. There is no elaboration as to the functional significance of this domain. Presumably, the CKK domain is not able to anchor the MTs in any way, since it is the minimal domain necessary to bind the MT. So, how does this fit into their model?

The authors claim that the defects observed in *patronin* knockdown are due to excessive depolymerization by *klp10a*. However, the number of EB1 comets is the same. It is just comet directionality that is perturbed. This does not make sense with their hypothesis. If it is excessive depolymerization by *klp10a*, then there should be far fewer comets. It would make sense if *klp10a* were a cross-linking or sliding kinesin, but a depolymerising kinesin only having an effect on orientation and not number of EB1 comets seems bizarre. Why would misorientation be a result of excessive depolymerization activity?

The authors state: "As a result, dendritic MTs may collapses into short and unstable filaments, which no longer anchor within the dendrites, instead fall into the soma." How does a filament fall into the soma? Likely, the MT would just depolymerize entirely, or depolymerize to a point where another stabilizing MAP is located and regrow. The concentration of tubulin in most cells favors new MT growth, and perhaps this new growth could be oriented incorrectly and this is why there is mixed MT polarity. Maybe the absence of Patronin affects another MAP that is necessary for the orientation of the MTs. As the authors note, many proteins result in a mixed MT polarity upon knockdown. There are many different MAPs that could be perturbed upon Patronin knockdown, and therefore this may not be a direct effect of Patronin function, but an indirect effect via other MAP functions.

Finally, why would uniform MT orientation facilitate MT breakdown? This also seem correlative, and the papers they cite provide nothing but speculation as to why the relationship between MT orientation and the progress of pruning. In order to say uniform MT orientation is necessary for MT breakdown and subsequent pruning, the authors would need to test a number of MAPs that have varying effects on MT orientation and assess their effects on the pruning timecourse.

*Reviewer #3:*

In this manuscript by Wang et al., the authors report a role for the microtubule minus end protein Patronin/CAMSAP/PTRN-1 in dendrite pruning in the dda dendrites in *Drosophila*. In addition, they find a role for Patronin in maintaining minus end out microtubule polarity in these dendrites as well. The CKK domain of Patronin is necessary and sufficient for its role in pruning. Both of these functions appear to operate through the antagonism of the depolymerizing kinesin Klp10A as the phenotypes can be suppressed upon removal of Klp10A function. Overall the data are well presented and the conclusions well supported by the data.

Two big questions I am left with from this work are:

1) Why does the maintenance of minus end out microtubules in the dendrite allow for productive pruning? The authors speculate a bit about this in the Discussion, but a clearer connection would elevate the paper. In particular all of the phenotypic observations are made in steady state dendrites. The authors assume the two phenotype- maintenance of minus end out MTs and efficient dendritic pruning -but this is speculation. Would it be possible to better characterize MT polarity in both control and *patronin* mutants during the pruning process, i.e. between WP and 16h APF?

2) I am confused by the Patronin gain of function phenotype. If the model is "that Patronin promotes dendritic pruning via antagonizing KLP10A function", then my expectation would be that Patronin overexpression would not lead to pruning defects. This model is at least supported by the MT polarity phenotype upon Patronin overexpression (i.e. that minus end out MT polarity is perturbed), but again this does not fit so neatly into the model presented.

In some cases, the data presented suggest precocious pruning in some mutant conditions. For example, Figure 7G and H: In the Klp10A OE, there appears to be fewer branches at the WP stage as compared to controls. Is this the case?

Where does Patronin localize in the neurons at 1) steady state, and 2) during pruning?

Minor Points:

Figure 4 legend is out of order in its description of the panels, making it hard to follow

Figure 4F-H: where are the dendrites here? Aren't they supposed to be labelled with curly brackets as is indicated in the legend? Also, the Kin-B-gal is very hard to see. What are the arrowheads pointing to?

In many places problems with the writing make it difficult to interpret what the authors are saying. For example, the description of Figure 5 in subsection “Patronin is required for the minus-end-out orientation of dendritic MTs in ddaC neurons” is particularly confusing. The data presented in the figure are clear, but the wording muddies the authors' intent.

In the same section: "over 80% of anterograde branches moved away from the soma"-isn't this the definition of anterograde?

Minor corrections:

Results section, first paragraph: mutant clones.

Various points throughout the Results section: On average.

Subsection “Overexpression of Patronin causes dendrite pruning defects in ddaC neurons”, first paragraph, final sentence: as a.

Subsection “Attenuation of Klp10A, a kinesin-13 MT depolymerase, suppresses the *patronin* phenotype in MT orientation in ddaC dendrites”: Previous studies or A previous study.

Discussion section, first paragraph: PTRN-1.

Figure 4D and H axis label "moving"

Figure 4: What is GFP labeling here? No explanation for this in figure legend or text.

[Editors' note: further revisions were suggested, as described below.]

Thank you for resubmitting your work entitled "Patronin governs minus-end-out orientation of dendritic microtubules to promote dendrite pruning in *Drosophila*" for further consideration at *eLife*. Your revised article has been favorably evaluated by Anna Akhmanova (Senior Editor), a Reviewing Editor, and two reviewers.

The manuscript has been improved but there are some remaining, as outlined below:

Please address the remaining comments by including the katanin knockdown data if you have it. Please also expand discussions on the issue raised by reviewer 1. Upon receiving your revision, I will make a decision without sending it to the reviewers.

*Reviewer #1:*

The revisions have strengthened several points, but other key questions remain open. My overall impression is still that the manuscript describes a very interesting observation, although the underlying mechanism is still unclear.

– The connection between MT polarity (rather than stability) and pruning is now stronger. However, how polarity actually affects pruning is not explored.

– I couldn't find the phenotype of the Katanin single knock down, which makes interpreting the double knock down with Patronin impossible. I think that this is a technical point that has to be addressed or this experiment should be removed.

– The conclusion that Patronin loss and gain of function lead to pruning defects in the same way has been strengthened by the revisions. It is still unclear to me how this actually fits into the proposed mechanism (see below).

– The revisions haven't clarified the mechanism through which Patronin affects polarity and not stability. There is some disagreement between the reduced Futch staining and the normal number of plus-end comets. The authors' model about Klp10A affecting only minus ends does not explain why it would stop depolymerizing before overtaking the entire polymer, or how it distinguishes the minus ends of minus-end-out vs those of plus-end-out MTs. As a minor point, it is possible that short MTs would turn around in the axon or dendrite as they suggest, but the reference cited (del Castillo et al) describes MT sliding by dynein and kinesin-1 and it is not clear why the authors chose it to support their model.

---

## [Author Response]

Major concerns:1) Reviewers found that the causality between MT polarity defects and pruning is lacking. We suggest you to examine other mutants with MT polarity defects like cnn, APC to see if they have pruning phenotypes.

We have now examined the possible role of cnn and APC1/2 (known regulators of dendritic MT polarity) in dendrite pruning. We observed dendrite severing defects in 52% of cnnhk21 mutant ddaC neurons (Figure 8—figure supplement 3A). Likewise, APC1Q8, APC2N175K double mutant ddaC neurons also exhibited dendrite severing defects (60%, Figure 8—figure supplement 3B). Thus, our new data further support the conclusion that proper dendritic MT polarity is important for dendrite pruning. We have now included these new data in the revised Figure 8—figure supplement 3A-B and text (subsection “A high correlation between MT orientation and dendrite-specific pruning in ddaC sensory neurons”).

2) The authors hypothesized that the defects observed in Patronin knockdown are due to excessive depolymerization by klp10a. However, the number of EB1 comets is the same. It is just comet directionality that is perturbed. If the excessive depolymerization by klp10a is the cause of the phenotype, then there should be far fewer comets. It would make sense if klp10a were a cross-linking or sliding kinesin, but a depolymerising kinesin only having an effect on orientation and not number of EB1 comets seems bizarre.

It was indeed unexpected that the number of EB1 comets remained unchanged in dendrites upon *patronin* RNAi knockdown. Two co-authors of this manuscript have repeated the experiment and found the same result (revised Figure 5D). Moreover, Klp10A overexpression did not cause a signification reduction in the number of EB1-GFP comets in the dendrites (Figure 8I) but caused the MT polarity defect in the dendrites (Figure 8G-H) (subsection “Klp10A overexpression phenocopies *patronin* knockdown in dendrite pruning and dendritic MT orientation”), resembling the *patronin* RNAi knockdown (Figure 5D). These data imply that *patronin* RNAi or Klp10A overexpression might result in excessive MT depolymerization primarily at the MT minus ends, rather than affect the number of EB1-GFP comets that mark MT growth at the MT plus ends. Consistent with this notion, it was reported that in Patronin-deficient *Drosophila* S2 cells, minus end depolymerization often halted when it reached the EB1-enriched MT plus end tips, indicating that +TIP proteins including EB1 might resist continued minus-end depolymerization (Goodwin and Vale, 2010). On the other hand, in worm touch neurons, depletion of the *patronin* homolog leads to an increase in the number of EB1 comets (Chuang et al., 2014). We have now included the new data in the revised Figure 8F-K and text (subsection “Klp10A overexpression phenocopies *patronin* knockdown in dendrite pruning and dendritic MT orientation”).

Why would misorientation be a result of excessive depolymerization activity? This is an important question that needs to be addressed because it gets at the fundamental question on the relationship between MT stability and polarity.

In the absence of Patronin, dendritic MTs might be depolymerized into short filaments from their minus ends mediated by excessive depolymerization activity of Klp10A and/or other MT severing factors. Microtubule depolymerising or severing factors have been observed to depolymerize MTs into short filaments (McNally F. et al., Cell 1993). Short MT filaments, which was proposed to be re-oriented in either plus-end-out or minus-end-out direction with equal probability (del Castillo et al., 2015), can serve as seeds for MT growth, resulting in a mixed MT polarity in the dendrites of *patronin* mutant neurons. We have now included this discussion in the revised manuscript (subsection “Patronin regulates the minus-end-out orientation of dendritic MTs in ddaC dendrites”).

3) Genetic interaction between Patronin and Katanin kat60L should be addressed.

A previous study reported that the putative MT severing factor Kat-60L1 is required for dendrite pruning (Lee et al., 2009). We have now conducted genetic interactions between *patronin* and *kat-60L1*. When Kat-60L1 was depleted via two RNAi lines in the *patronin* RNAi background, we did not observe any enhancement or suppression between *patronin* and *kat-60L1* in dendrite pruning. Thus, this new data indicates no genetic interaction between *patronin* and *kat-60L1*. We have now included the new data in the revised Figure 7—figure supplement 1C) and text (subsection “Patronin promotes dendrite pruning by orienting uniform minus-end-out MT arrays in dendrites”).

4) Please address whether Patronin overexpression affects dendrite pruning through the same mechanism as the loss-of-function mutant. This overexpression effect is not consistent with the model provided.

To address whether Patronin overexpression, like the loss-of-function mutant, also affects dendrite pruning through excessive Klp10A activity, we have knocked down *klp10A* in Patronin-overexpressing ddaC neurons. Knockdown of *klp10A*, via two independent RNAi lines, significantly suppressed the dendrite pruning defects in Patronin-overexpressing ddaC neurons (Figure 7F-J). Moreover, knockdown of *klp10A* also significantly suppressed the MT orientation defect in the dendrites of Patronin-overexpressing ddaC neurons (Figure 7K-N). Therefore, these results suggest that both gain of *patronin* function and loss of *patronin* function affects dendrite pruning at least partially by upregulation of Klp10A function. We have now included the new data in the revised Figure 7F-N and text (subsection “Patronin promotes dendrite pruning by orienting uniform minus-end-out MT arrays in dendrites”).

5) The CKK domain of Patronin is necessary and sufficient for pruning, which seems to be a surprising result considering that it is the minimal domain necessary to bind the MT. How can one explain this result?

In the original manuscript, we show that CKK-deleted Patronin variant (PatroninΔCKK) failed to rescue the dendrite pruning defects in *patronin*^c9-c5^ ddaC neurons, whereas the expression of the CKK domain significantly rescued *patronin*^c9-c5^-associated dendrite pruning defects, suggesting that the CKK domain is necessary and sufficient for Patronin’s function during dendrite pruning. We have now conducted additional experiments to elaborate the functional significance of the CKK domain. First, we confirmed that the expression of the CKK domain alone significantly rescued the dendrite pruning defects in the *patronin* RNAi background (Figure 8—figure supplement 2A). Second, the expression of the CKK domain almost fully rescued Nod-lacZ distribution as well as retrograde EB1-GFP comets in the dendrites of *patronin* RNAi ddaC neurons (Figure 8—figure supplement 2B-C). Third, the expression of the CKK domain significantly suppressed the dendrite pruning defects in Klp10A-overexpressing ddaC neurons (Figure 8—figure supplement 2D), suggesting that the CKK domain is able to antagonize Klp10A’s function during dendrite pruning. Thus, multiple lines of evidence demonstrate that the CKK domain is important for Patronin to govern minus-end-out MT orientation in dendrites as well as dendrite pruning. Thus, we propose the model in which via its CKK domain, Patronin is essential for the establishment of minus-end-out MTs in the dendrites of ddaC neurons. We have now included these new data in the revised manuscript (subsection “Klp10A overexpression phenocopies *patronin* knockdown in dendrite pruning and dendritic MT orientation”).

We have also tuned down the conclusion as “The CKK domain of Patronin is important for Patronin to govern minus-end-out MT orientation in dendrites as well as dendrite pruning” in the revised manuscript.

We would also like to invite you to respond to the minor points that are listed in the individual reviews.Separate reviews (please respond to each point):

Reviewer #1:

This is a very interesting paper that describes an unexpected role for Patronin in controlling dendrite MT polarity and pruning in *Drosophila*. The experiments are overall convincing and are properly controlled and the topic and scope are fit for eLife.My main concern is that although this paper makes a very interesting connection between microtubule stability, polarity and pruning, it is not clear from the experiments how stabilization of MTs favors minus-end-out orientation, and whether it is polarity or stability that is important for pruning. This latter point could be addressed by examining some of the other mutants that affect MT polarity (cnn, APC, EB or others) or stability (Futch, Tau or others). by using genetic interactions the authors could place the Patronin-KLP10 module in the context of what we already know about MT regulation in this system.

We have now examined the possible role of cnn and APC1/2 (regulators of dendritic MT polarity) as well as futsch and tau (regulators of MT stability) in dendrite pruning. We observed dendrite severing defects in 52% of cnnhk21 mutant ddaC neurons (Figure 8—figure supplement 3A). Likewise, APC1Q8, APC2N175K double mutant ddaC neurons also exhibited dendrite severing defects (60%, Figure 8—figure supplement 3B). In contrast, no dendrite pruning defect was observed in fustchN94 or tau RNAi ddaC neurons (Figure 8—figure supplement 3C-D). These new data, together with our *patronin* results, further indicate that proper dendritic MT polarity is important for dendrite pruning. We have now included the new data in the revised Figure 8—figure supplement 3 and text (subsection “A high correlation between MT orientation and dendrite-specific pruning in ddaC sensory neurons”).

Previous studies reported that the putative MT severing factor Kat-60L1 and the MT-associated protein Tau are involved in dendrite pruning (Lee et al., 2009; Herzmann et al., 2017). We have now conducted genetic interactions between *patronin* and kat-60L1/tau. When either Kat-60L1 or Tau was knocked down via two RNAi lines in the *patronin* RNAi background, we did not observe any enhancement or suppression between *patronin* and kat-60L1/tau in dendrite pruning (Figure 7—figure supplement 1C-D). Thus, these results indicate that MT polarity but not MT stability is important for dendrite pruning. These new results, together with our original result that the *patronin* mutant phenotypes were fully rescued by klp10A RNAi knockdown, support the conclusion that the Patronin-Klp10A module is a novel regulatory pathway that governs MT polarity as well as dendrite pruning. We have now included the new data in the revised Figure 7—figure supplement 1C-D and text (subsection “Patronin promotes dendrite pruning by orienting uniform minus-end-out MT arrays in dendrites”).

We propose that in wild-type ddaC neurons, Patronin likely associates with non-centrosomal MT minus-ends and stabilizes them against kinesin-13-mediated MT depolymerization in the dendrites. Via MT guidance or sliding by plus-end motors kinesins, growing MTs might be oriented in a minus-end-out manner in the dendrites (Yan et al., 2013; Mattie et al., 2010). In the absence of Patronin, dendritic MTs might be depolymerized into short filaments from their minus ends mediated by excessive depolymerization activity of Klp10A and/or other MT severing factors. Microtubule depolymerising or severing factors have been observed to depolymerize MTs into short filaments (McNally et al., Cell 1993). Short MT filaments, which was proposed to be re-oriented in either plus-end-out or minus-end-out direction with equal probability (del Castillo et al., 2015), can serve as seeds for MT growth, resulting in a mixed MT polarity in the dendrites of *patronin* mutant neurons. We have now included this discussion in the revised manuscript (subsection “Patronin regulates the minus-end-out orientation of dendritic MTs in ddaC dendrites”).

My second reservation is that I am not convinced that Patronin overexpression affects dendrite pruning through the same mechanism as the loss-of-function as the paper seems to imply. This is because OE of the CKK domain has no effect and because loss of KLP10 (which should be equivalent to Patronin overexpression) has no phenotype. The authors should clearly discuss the possible distinct mechanisms of Patronin loss and gain of function.

To address whether Patronin overexpression, like the loss-of-function mutant, also affects dendrite pruning through excessive Klp10A activity, we have knocked down klp10A in Patronin-overexpressing ddaC neurons. Knockdown of klp10A, via two independent RNAi lines, significantly suppressed the dendrite pruning defects in Patronin-overexpressing ddaC neurons (Figure 7F-J). Moreover, knockdown of klp10A also significantly suppressed the MT orientation defect in the dendrites of Patronin-overexpressing ddaC neurons (Figure 7K-N). Therefore, these results suggest that both gain of *patronin* function and loss of *patronin* function affects dendrite pruning at least partially by upregulation of Klp10A function. We have now included the new data in the revised Figure 7F-N and text (subsection “Patronin promotes dendrite pruning by orienting uniform minus-end-out MT arrays in dendrites”).

Minor Comments:– Please show more representative kymographs for the EB movies (i.e that show that most of the time the phenotype is not a complete reversal), and move the quantification from the supplementary figure to main figure.

We have now showed the representative kymographs for EB1 movies and moved the quantification from the supplemental Figure 5 to the main Figure 5M in the revised manuscript. Please note that in the revised Figures 5-6, the new images were acquired at different experimental conditions, compared with images shown in our initial manuscript. We previously reared the larva in the grape-juice food to obtain brighter EB1-GFP signals. Due to the poor nutrition, those larvae appeared less healthy with reduced eclosion rate, likely due to cellular stress. To improve the growth condition, we have now reared the larvae on normal cornmeal food without yeast supplement (live yeast often generates auto-fluorescent signals). In the new condition, the larvae were much healthier and survived to the adulthood. Their ddaC neurons also showed detectable EB1-GFP comets, however, with fewer comet number than those reared in the grape-juice food. Importantly, two co-authors of this manuscript independently obtained similar results on EB1 comet movement and MT orientation from both *patronin* RNAi and Patronin-overexpressing neurons in larvae on the new growth condition. Therefore, we took this opportunity to replace the previous EB1 images (the original Figures 5-6) with the new images that are representative and obtained from larvae with an improved growth condition. We have included this information in the revised Materials and methods.

– Please quantify MT growth speed and run length for EB movies. This could help to understand whether there are different properties to the minus-end-out versus plus-end-out nucleation.

We have now included the quantification results of MT growth speed and tracking length for EB movies in the revised Figures 5-6 as well as text (subsection “Patronin is required for the minus-end-out orientation of dendritic MTs in ddaC neurons”).

– Is there any effect on axonal MT polarity (EB movies) in Patronin loss of function or KLP10 OE?

We have now examined EB1 movement in the axons of *patronin* RNAi ddaC neurons. Compared with 4% of retrograde EB1-GFP comets (predominantly in a plus-end-out orientation) in the axons of wild-type neurons, *patronin* knockdown led to 41% of retrograde EB1-GFP comets in the axons (Figure 5—figure supplement 1C), suggesting a severe defect in the plus-end-out orientation of their axonal MTs. Thus, *patronin* plays an important role in both dendritic and axonal MT polarity. We have now included the new data in the revised Figure 5—figure supplement 1C and text (subsection “Patronin is required for the minus-end-out orientation of dendritic MTs in ddaC neurons”).

– Please add tubulin staining to support the conclusion that overall MT levels are not changed.

In the original manuscript, we did not conduct tubulin staining and did not claim that overall MT levels are not changed. We have now detected the overall polymerized MT levels in the dendrites of *patronin* RNAi ddaC neurons using the antibody 22C10 against Futsch, a microtubule-associated protein. Our quantification data indicate that the intensity of overall MT levels was significantly reduced in the dendrites of *patronin* RNAi ddaC neurons, compared with the controls (Figure 5—figure supplement 1B). This finding is consistent with the previous reports that MT density is reduced upon the depletion of Patronin or its mammalian homologs (Goodwin and Vale, 2010; Yau et al., 2014). Thus, our new data indicate that Patronin regulates both MT polarity and levels in the dendrites. We have now included the new data in the revised Figure 5—figure supplement 1B and text (subsection “Patronin is required for the minus-end-out orientation of dendritic MTs in ddaC neurons”).

– What happens at later time-points? Do the mutant dendrites eventually "catch-up"?

We have now examined the dendrite pruning process at two late time points, 24 h APF and 32 h APF. The larval dendrites of *patronin* RNAi ddaC neurons were largely removed by 32 h APF (Figure 1—figure supplement 1B), presumably due to large-scale apoptosis and migration of the dorsal abdominal epidermis, on which ddaC neurons arborize their larval dendrites (William and Truman, 2005a). This is similar to the final pruning of EcRDN-overexpressing ddaC/D/E neurons at 30 h APF (Williams and Truman, 2005; Kirilly et al., 2009; Kirilly et al., 2011). We have now included the new data in the revised Figure 1—figure supplement 1B and text (subsection “Patronin is required for dendrite pruning and arborization in ddaC neurons”).

– I was confused by the fact that the Patronin phenotype is 100% penetrant with the overexpressed motor fusions but only partial with the EB comets. Does this mean that these motor constructs "read" something besides MT polarity? Please comment on this.

The penetrance difference was caused by the different quantification methods. For the Nod-β-Gal phenotype, we quantified the percentage of neurons with aberrant Nod-β-Gal distribution, whereas for the EB1-GFP phenotype, we quantified the percentage of comets with anterograde movement. If we quantify the percentage of neurons with anterograde EB1-GFP movement, we have now observed 93% of *patronin* RNAi neurons showing abnormal EB1-GFP movement direction in the dendrites of *patronin* RNAi neurons. This penetrance of EB1-GFP phenotype (93%) is comparable with Nod-β-lacZ penetrance (100%). In addition, EB1-GFP was expressed at a low level under the control of the weaker Gal4 driver Gal44-77 in order to visualize its comets, whereas Nod-lacZ was driven by the strong driver ppk-Gal4. Thus, the efficiency of *patronin* RNAi knockdown using Gal44-77 is likely lower than that using ppk-Gal4, leading to the partial EB1-GFP phenotype.

Reviewer #2:

In their manuscript, Wang, Rui and Yu investigate the role of the microtubule (MT) minus-end binding protein, Patronin, in *Drosophila* dendritic arborization neurons. The open questions the authors sought to address were: 1) the role of Patronin in neuronal development and pruning, and 2) the contribution of the minus-end out MT orientation to dendrite pruning. Both knockdown and overexpression of Patronin inhibits the normal timecourse of dendritic pruning in the Drosophila pupal dendritic arborization neurons. They find that Patronin is necessary for the maintenance of MT polarity in dendrites. Instead of microtubules being predominantly minus-end out, in Patronin knockdown or mutant neurons, there is a mixture of minus- and plus-end out MTs. Inhibiting the MT depolymerizing kinesein, klp10A (kinesin-13), in this background restores the polarity of the MTs, indicating that in the absence of patronin, klp10A somehow contributes to this mixed MT polarity. Inhibiting klp10a also rescues the dendritic pruning defects observed in the absence of Patronin. The overall message of the paper is that Patronin antagonizes klp10a activity normally to maintain uniform MT minus-end out polarity during dendritic maintenance and pruning. In the absence of patronin, klp10A somehow functions in disrupting the uniform polarity of microtubules in the dendrites, which in turn disrupts dendritic pruning. The data are very nice and the results are clear; however, the model is incomplete. The effect of Patronin and klp10a on MTs are correlative with the pruning defects, not causative, and there can be a wide variety of other MAPs (stabilizers, severing proteins, etc.) whose functions are directly or indirectly disrupted upon Patronin or klp10a knockdown. These other possibilities are completely ignored, making the model disjointed and speculative.

We have now examined the possible role of cnn and APC1/2 (regulators of dendritic MT polarity) as well as futsch and tau (MT stabilizers) in dendrite pruning. We observed dendrite severing defects in 52% of cnnhk21 mutant ddaC neurons (Figure 8—figure supplement 3A). Likewise, APC1Q8, APC2N175K double mutant ddaC neurons also exhibited dendrite severing defects (60%, Figure 8—figure supplement 3B). In contrast, no dendrite pruning defect was observed in fustchN94 or tau RNAi ddaC neurons (Figure 8—figure supplement 3C-D). These new data, together with our *patronin* results, further indicate that proper dendritic MT polarity is important for dendrite pruning. We have now included the new data in the revised Figure 8—figure supplement 3 and text (subsection “A high correlation between MT orientation and dendrite-specific pruning in ddaC sensory neurons”).

Previous studies reported that the putative MT severing factor Kat-60L1 and the MT-associated protein Tau are involved in dendrite pruning (Lee et al., 2009; Herzmann et al., 2017). We have now conducted genetic interactions between *patronin* and kat-60L1/tau. When either Kat-60L1 or Tau was knocked down via two RNAi lines in the *patronin* RNAi background, we did not observe any enhancement or suppression between *patronin* and kat-60L1/tau in dendrite pruning. Thus, these data suggest that Tau or Kat-60L1 functions are unlikely altered upon Patronin RNAi knockdown. We have now included the new data in the revised Figure 7—figure supplement 1C-D and text (subsection “Patronin promotes dendrite pruning by orienting uniform minus-end-out MT arrays in

dendrites”).

The authors do not mention the direct interaction between Patronin and the MT severing enzyme, katanin (Jiang et al., 2014 and Jiang et al., 2018, the latter of which was not even cited).

We have now cited Jiang et al., 2018, in the revised manuscript (subsection “Patronin promotes dendrite pruning by orienting uniform minus-end-out MT arrays in dendrites”).

The authors do mention that katanin itself does not have a role in pruning, but katanin p60-like does based on prior work. It seems important to look at the genetic interaction between Patronin and Kat60L1 as well, before constructing an entire model around Patronin and klp10a.

Given the direct interaction between mammalian Patronin (CAMSAPs) and katanin (Jiang et al., 2014 and 2018), we have now examined the potential genetic interactions between *patronin* and kat-60 in dendrite pruning. RNAi Knockdown of kat-60, via two distinct RNAi lines, did not enhance or suppress the pruning defects in *patronin* RNAi ddaC neurons (Figure 7—figure supplement 1B). Likewise, although Kat-60L1 was reported to be required for dendrite pruning (Lee et al., 2009), neither did double knockdown of kat-60L1 and *patronin* exhibit any enhancement or suppression in the dendrite pruning defects (Figure 7—figure supplement 1C). Thus, these data indicate no genetic interaction between *patronin* and kat-60/kat-60L1. We have now included the new data in the revised Figure 7—figure supplement 1B-C and text (subsection “Patronin promotes dendrite pruning by orienting uniform minus-end-out MT arrays in dendrites”).

The authors state many times that Patronin may stabilize uniform minus-end out MTs in the dendrites, but do not explain why and are unable to get at a mechanism in their paper. If the authors think that Patronin anchors MT minus ends in some way in the dendrites, then the CKK results do not make sense. The CKK domain of Patronin is necessary and sufficient for pruning. There is no elaboration as to the functional significance of this domain. Presumably, the CKK domain is not able to anchor the MTs in any way, since it is the minimal domain necessary to bind the MT. So, how does this fit into their model?

In the original manuscript, we show that CKK-deleted Patronin variant (PatroninΔCKK) failed to rescue the dendrite pruning defects in *patronin*^c9-c5^ ddaC neurons, whereas the expression of the CKK domain significantly rescued *patronin*^c9-c5^-associated dendrite pruning defects, suggesting that the CKK domain is necessary and sufficient for Patronin’s function during dendrite pruning. We have now conducted additional experiments to elaborate the functional significance of the CKK domain. First, we confirmed that the expression of the CKK domain alone significantly rescued the dendrite pruning defects in the *patronin* RNAi background (Figure 8—figure supplement 2A). Second, the expression of the CKK domain almost fully rescued Nod-lacZ distribution as well as retrograde EB1-GFP comets in the dendrites of *patronin* RNAi ddaC neurons (Figure 8—figure supplement 2B-C). Third, the expression of the CKK domain significantly suppressed the dendrite pruning defects in Klp10A-overexpressing ddaC neurons (Figure 8—figure supplement 2D), suggesting that the CKK domain is able to antagonize Klp10A’s function during dendrite pruning. Thus, multiple lines of evidence demonstrate that the CKK domain is important for Patronin to govern minus-end-out MT orientation in dendrites as well as dendrite pruning. Thus, we propose the model in which via its CKK domain, Patronin is essential for the establishment of minus-end-out MTs in the dendrites. We have now included these new data in the revised Figure 8—figure supplement 2 and text (subsection “Klp10A overexpression phenocopies *patronin* knockdown in dendrite pruning and dendritic MT orientation”).

We have also tuned down the conclusion as “The CKK domain of Patronin is important for Patronin to govern minus-end-out MT orientation in dendrites as well as dendrite pruning” in the revised manuscript. This revised conclusion fits into the model better.

The authors claim that the defects observed in patronin knockdown are due to excessive depolymerization by klp10a. However, the number of EB1 comets is the same. It is just comet directionality that is perturbed. This does not make sense with their hypothesis. If it is excessive depolymerization by klp10a, then there should be far fewer comets. It would make sense if klp10a were a cross-linking or sliding kinesin, but a depolymerising kinesin only having an effect on orientation and not number of EB1 comets seems bizarre.

It was indeed unexpected that the number of EB1 comets remained unchanged in dendrites upon *patronin* RNAi knockdown. Two co-authors of this manuscript have repeated the experiment and found the same result (revised Figure 5D). Moreover, Klp10A overexpression did not cause a signification reduction in the number of EB1-GFP comets in the dendrites (Figure 8I) but caused the MT polarity defect in the dendrites (Figure 8G-H) (subsection “Klp10A overexpression phenocopies *patronin* knockdown in dendrite pruning and dendritic MT orientation”), resembling the *patronin* RNAi knockdown (Figure 5D). These data imply that *patronin* RNAi or Klp10A overexpression might result in excessive MT depolymerization primarily at the MT minus ends, rather than affect the number of EB1-GFP comets that mark MT growth at the MT plus ends. Consistent with this notion, it was reported that in Patronin-deficient *Drosophila* S2 cells, minus end depolymerization often halted when it reached the EB1-enriched MT plus end tips, indicating that +TIP proteins including EB1 might resist continued minus-end depolymerization (Goodwin and Vale, 2010). On the other hand, in worm touch neurons, depletion of the *patronin* homolog leads to an increase in the number of EB comets (Chuang, et al., 2014). We have now included the new data in the revised Figure 8F-K and text (subsection “Klp10A overexpression phenocopies *patronin* knockdown in dendrite pruning and dendritic MT orientation”).

Why would misorientation be a result of excessive depolymerization activity?

In the absence of Patronin, dendritic MTs might be depolymerized into short filaments from their minus ends mediated by excessive depolymerization activity of Klp10A and/or other MT severing factors. Microtubule depolymerising or severing factors have been observed to depolymerize MTs into short filaments (McNally F. et al., Cell 1993). Short MT filaments, which was proposed to be re-oriented in either plus-end-out or minus-end-out direction with equal probability (del Castillo et al., 2015), can serve as seeds for MT growth, resulting in a mixed MT polarity in the dendrites of *patronin* mutant neurons. We have now included this discussion in the revised manuscript (subsection “Patronin regulates the minus-end-out orientation of dendritic MTs in ddaC dendrites”).

The authors state: "As a result, dendritic MTs may collapses into short and unstable filaments, which no longer anchor within the dendrites, instead fall into the soma." How does a filament fall into the soma? Likely, the MT would just depolymerize entirely, or depolymerize to a point where another stabilizing MAP is located and regrow. The concentration of tubulin in most cells favors new MT growth, and perhaps this new growth could be oriented incorrectly and this is why there is mixed MT polarity.

We thank the reviewer for the helpful comment on our hypothesis. We have rephrased the sentences as “In the absence of Patronin, dendritic MTs might be depolymerized into short filaments from their minus ends mediated by excessive depolymerization activity of Klp10A and/or other MT severing factors. Microtubule depolymerising or severing factors have been observed to depolymerize MTs into short filaments (McNally F. et al., Cell 1993). Short MT filaments, which was proposed to be re-oriented in either plus-end-out or minus-end-out direction with equal probability (del Castillo et al., 2015), can serve as seeds for MT growth, resulting in a mixed MT polarity in the dendrites of *patronin* mutant neurons.” in the revised Discussion (subsection “Patronin regulates the minus-end-out orientation of dendritic MTs in ddaC dendrites”).

Maybe the absence of Patronin affects another MAP that is necessary for the orientation of the MTs. As the authors note, many proteins result in a mixed MT polarity upon knockdown. There are many different MAPs that could be perturbed upon Patronin knockdown, and therefore this may not be a direct effect of Patronin function, but an indirect effect via other MAP functions.

We have now examined the possible role of two important MAPs, namely Futsch and Tau in dendrite pruning. No dendrite pruning defect was observed in fustchN94 or tau RNAi ddaC neurons (Figure 8—figure supplement 3C-D). Moreover, we have also conducted genetic interactions between *patronin* and kat-60L1/tau. When either Kat-60L1 or Tau was knocked down via two RNAi lines in the *patronin* RNAi background, we did not observe any enhancement or suppression between *patronin* and kat-60L1/tau in dendrite pruning (Figure 7—figure supplement 1C-D). Formally, we cannot exclude the possibility that the MT orientation phenotypes in *patronin* mutant neurons might be caused by impaired functions of multiple MAPs. In our current study, we focus mainly on the Patronin-Klp10A module in which Patronin regulates dendritic MT orientation at least partially by antagonizing the MT depolymerizing kinesin Klp10A. We have now included these new data in the revised figures (Figure 7—figure supplement 1C-D, Figure 8—figure supplement 3C-D) and text (subsections “Patronin promotes dendrite pruning by orienting uniform minus-end-out MT arrays in dendrites” and “A high correlation between MT orientation and dendrite-specific pruning in ddaC sensory neurons”).

Finally, why would uniform MT orientation facilitate MT breakdown? This also seem correlative, and the papers they cite provide nothing but speculation as to why the relationship between MT orientation and the progress of pruning. In order to say uniform MT orientation is necessary for MT breakdown and subsequent pruning, the authors would need to test a number of MAPs that have varying effects on MT orientation and assess their effects on the pruning timecourse.

We have now examined the possible role of cnn and APC1/2 (known regulators of dendritic MT polarity) in dendrite pruning. We observed dendrite severing defects in 52% of cnnhk21 mutant ddaC neurons (Figure 8—figure supplement 3A). Likewise, APC1Q8, APC2N175K double mutant ddaC neurons also exhibited dendrite severing defects (60%, Figure 8—figure supplement 3B). In addition, both kinesin-1 and kinesin-2 mutant ddaC neurons with mixed dendritic MT orientations also exhibited dendrite pruning defects (Mattie et al., 2010; Herzmann et al., 2018). Thus, multiple lines of evidence demonstrate that uniform dendritic MT orientation is important for dendrite pruning. We have now included the new data in the revised Figure 8—figure supplement 3A-B and text (subsection “A high correlation between MT orientation and dendrite-specific pruning in ddaC sensory neurons”).

Reviewer #3:

In this manuscript by Wang et al., the authors report a role for the microtubule minus end protein Patronin/CAMSAP/PTRN-1 in dendrite pruning in the dda dendrites in *Drosophila*. In addition, they find a role for Patronin in maintaining minus end out microtubule polarity in these dendrites as well. The CKK domain of Patronin is necessary and sufficient for its role in pruning. Both of these functions appear to operate through the antagonism of the depolymerizing kinesin Klp10A as the phenotypes can be suppressed upon removal of Klp10A function. Overall the data are well presented and the conclusions well supported by the data.Two big questions I am left with from this work are:1) Why does the maintenance of minus end out microtubules in the dendrite allow for productive pruning? The authors speculate a bit about this in the Discussion, but a clearer connection would elevate the paper. In particular all of the phenotypic observations are made in steady state dendrites. The authors assume the two phenotype- maintenance of minus end out MTs and efficient dendritic pruning -but this is speculation. Would it be possible to better characterize MT polarity in both control and patronin mutants during the pruning process, i.e. between WP and 16h APF?

To further substantiate the link between microtubule polarity and dendrite pruning, we have now examined the possible role of cnn and APC1/2 (known regulators of dendritic MT polarity) in dendrite pruning. We observed dendrite severing defects in 52% of cnnhk21 mutant ddaC neurons (Figure 8—figure supplement 3A). Likewise, APC1Q8, APC2N175K double mutant ddaC neurons also exhibited dendrite severing defects (60%, Figure 8—figure supplement 3B). In addition, both kinesin-1 and kinesin-2 mutant ddaC neurons with mixed dendritic MT orientations also exhibited dendrite pruning defects (Mattie et al., 2010; Herzmann et al., 2018). Thus, multiple lines of evidence demonstrate that uniform dendritic MT orientation is important for dendrite pruning. We have now included the new data in the revised Figure 8—figure supplement 3A-B and text (subsection “A high correlation between MT orientation and dendrite-specific pruning in ddaC sensory neurons”).

We have also attempted to examine dendritic MT polarity in both control and *patronin* RNAi ddaC neurons at 6 h APF before the onset of dendrite pruning. Unfortunately, due to their opaque brown cases, the pupae showed undetectable EB1-GFP signals in ddaC neurons at this prepupal stage. Those pupal case cannot be peeled at this stage. To date, no lab in the field has successfully reported EB1-GFP imaging during the pruning process due to these technical limitations.

2) I am confused by the Patronin gain of function phenotype. If the model is "that Patronin promotes dendritic pruning via antagonizing KLP10A function", then my expectation would be that Patronin overexpression would not lead to pruning defects. This model is at least supported by the MT polarity phenotype upon Patronin overexpression (i.e. that minus end out MT polarity is perturbed), but again this does not fit so neatly into the model presented.

In the original version, our data suggest that Patronin overexpression may behave as a dominant negative, as it phenocopied *patronin* mutants in terms of dendrite pruning and dendritic MT orientation. To further substantiate this point, we have now knocked down *klp10A* in Patronin-overexpressing ddaC neurons. knockdown of *klp10A*, via two independent RNAi lines, significantly suppressed the dendrite pruning defects in Patronin-overexpressing ddaC neurons (Figure 7F-J). Moreover, knockdown of *klp10A* also significantly suppressed the MT orientation defect in the dendrites of Patronin-overexpressing ddaC neurons (Figure 7K-N). These results suggest that both gain of *patronin* function and loss of *patronin* function affects dendrite pruning at least partially by upregulation of Klp10A function. We have now included the new data in the revised Figure 7F-N and text (subsection “Patronin promotes dendrite pruning by orienting uniform minus-end-out MT arrays in dendrites”).

In some cases, the data presented suggest precocious pruning in some mutant conditions. For example, Figure 7G and H: In the Klp10A OE, there appears to be fewer branches at the WP stage as compared to controls. Is this the case?

Klp10A overexpression led to simplified dendrite arbors at the WP stage, which is the dendrite arborization defect. These simplified dendrite arbors persisted and remained attached to the soma at 16 h APF, indicative of dendrite pruning defects, but not the precocious dendrite pruning.

Where does Patronin localize in the neurons at 1) steady state, and 2) during pruning?

We have now conducted immuno-staining using anti-Patronin antibody at the larval (96 h AEL) and white pupal stages. Patronin localized uniformly to dendrites, axons and soma at both larval and pupal stages.We have now included the new data in the revised Figure 4—figure supplement 1B and text (subsection “Patronin is required for proper distribution of dendritic and axonal MT markers”).

Minor Points:Figure 4 legend is out of order in its description of the panels, making it hard to follow

We have now corrected the order in the description of the panels in the revised legends.

Figure 4F-H: where are the dendrites here? Aren't they supposed to be labeld with curly brackets as is indicated in the legend? Also, the Kin-B-gal is very hard to see. What are the arrowheads pointing to?

We have now added the curly brackets to highlight the dendrites in the revised Figure 4. We have also included the description of the arrowheads as follows, “Arrowheads point to ectopic Kin-β-gal aggregates in the dendrites” in the revised figure legends.

In many places problems with the writing make it difficult to interpret what the authors are saying. For example, the description of Figure 5 in subsection “Patronin is required for the minus-end-out orientation of dendritic MTs in ddaC neurons” is particularly confusing. The data presented in the figure are clear, but the wording muddies the authors' intent.

We have now rephrased the description of Figure 5 and made it more concise in the revised manuscript.

In the same section: "over 80% of anterograde branches moved away from the soma"-isn't this the definition of anterograde?

We have now deleted “anterograde” and modified the sentence as “In these mutant dendrite branches, over 80% of EB1-GFP comets moved away from the soma, suggesting a predominant plus-end-out MT pattern” in the revised text.

Minor corrections:Results section, first paragraph: mutant clones.

We have now made the change in the revised text.

Various points throughout the Results section: On average.

We have now made the corrections in the revised manuscript accordingly.

Subsection “Overexpression of Patronin causes dendrite pruning defects in ddaC neurons”, first paragraph, final sentence: as a

We have now made the correction in the revised text.

Subsection “Attenuation of Klp10A, a kinesin-13 MT depolymerase, suppresses the patronin phenotype in MT orientation in ddaC dendrites”: Previous studies or A previous study.

We have made the change in the revised manuscript.

Discussion section, first paragraph: PTRN-1.

We have now made the correction in the revised manuscript.

Figure 4D and H axis label "moving"

We have now changed to “moving” in the revised figures.

Figure 4: What is GFP labeling here? No explanation for this in figure legend or text

GFP labelling indicates the mCD8-GFP fusion protein. We have modified the original sentence to “Confocal images of ddaC neurons expressing mCD8-GFP, Nod-lacZ or Kin-lacZ and immunostained for β-galactosidase at wL3 stages.” in the legends of the revised Figure 4 as well as all other relevant figures.

[Editors' note: further revisions were suggested, as described below.]

Please address the remaining comments by including the katanin knockdown data if you have it. Please also expand discussions on the issue raised by reviewer 1. Upon receiving your revision, I will make a decision without sending it to the reviewers.

Reviewer #1:

The revisions have strengthened several points, but other key questions remain open. My overall impression is still that the manuscript describes a very interesting observation, although the underlying mechanism is still unclear.– The connection between MT polarity (rather than stability) and pruning is now stronger. However, how polarity actually affects pruning is not explored.

We thank the reviewer for the endorsement on our revised manuscript. In the current manuscript, we have, for the first time, demonstrated that the MT minus-end-bind protein Patronin plays an important role in regulating MT minus-end-out orientation in dendrites and thereby promoting dendrite pruning of sensory neurons. Thus, this study provides a firm basis for future investigations into several interesting questions, for example, how MT polarity affects pruning.

– I couldn't find the phenotype of the Katanin single knock down, which makes interpreting the double knock down with Patronin impossible. I think that this is a technical point that has to be addressed or this experiment should be removed.

Knockdown of *katanin (kat-60)*, via two distinct RNAi lines, did not affect normal dendrite pruning in ddaC neurons. We have now included these new data in the revised text (subsection “Patronin promotes dendrite pruning by orienting uniform minus-end-out MT arrays in dendrites”, paragraph two) and Figure 7—figure supplement 1B.

– The conclusion that Patronin loss and gain of function lead to pruning defects in the same way has been strengthened by the revisions. It is still unclear to me how this actually fits into the proposed mechanism (see below).– The revisions haven't clarified the mechanism through which Patronin affects polarity and not stability. There is some disagreement between the reduced Futch staining and the normal number of plus-end comets. The authors' model about Klp10A affecting only minus ends does not explain why it would stop depolymerizing before overtaking the entire polymer, or how it distinguishes the minus ends of minus-end-out vs those of plus-end-out MTs.

We have now included the following discussion in the revised Discussion part “*patronin* depletion might result in excessive MT depolymerization primarily at the MT minus ends. Consistent with this notion, it was reported that in Patronin-deficient *Drosophila* S2 cells, minus end depolymerization often halted when it reached the EB1-enriched MT plus end tips, indicating that +TIP proteins might resist continued minus-end depolymerization (Goodwin and Vale, 2010).”

As a minor point, it is possible that short MTs would turn around in the axon or dendrite as they suggest, but the reference cited (del Castillo et al) describes MT sliding by dynein and kinesin-1 and it is not clear why the authors chose it to support their model.

In the discussion part of the cited paper (del Castillo et al., 2015), the authors proposed and discussed the possibility that short MT fragments might be able to turn around in the axons or dendrites, although the main point of the paper focuses on MT sliding by dynein and kinesin-1.

We greatly appreciate the editors for kind support and advice. Following the editors’ advice, in the revised manuscript we have included the *katanin* knockdown data and also expanded the discussion on the issue raised by the reviewer #1.